# Competitiveness analysis of fresh tomatoes in Indonesia: Turning comparative advantage into competitive advantage

Saptana[1], Syahrul Ganda Sukmaya[2]*, Atika Dyah Perwita[3], Fadhila Dhia Malihah[4], Irwanda Wisnu Wardhana[1], Atika Dian Pitaloka[5], Shabrina Austin Ghaisani[6], Bambang Sayaka[7], Nyak Ilham[8], Elna Karmawati[9], Mewa Ariani[8], Sri Hery Susilowati[8], Sumaryanto[8], Handewi Purwati Saliem[8]

1 Research Center for Cooperative, Corporation, and People's Economy, National Research and Innovation Agency, Jakarta, Indonesia, 2 Faculty of Agriculture, Universitas Jenderal Soedirman, Purwokerto, Indonesia, 3 Faculty of Education and Teacher Training, Universitas Islam Negeri Walisongo Semarang, Semarang, Central Java, Indonesia, 4 Faculty of Economics and Business, Universitas Mercu Buana, Jakarta, Indonesia, 5 Master of Management and Business, SB IPB University, Bogor, Indonesia, 6 Faculty of Engineering, Engineering of Management, University of Melbourne, Melbourne, Australia, 7 Research Centre for Economics of Industry, Services, and Trade, National Research and Innovation Agency, Jakarta, Indonesia, 8 Research Center for Behavioral and Circular Economics, National Agency for Research and Innovation, Jakarta, Indonesia, 9 Research Center for Horticultural and Estate Crops, National Agency for Research and Innovation, Jakarta, Indonesia

* syahrul.ganda@unsoed.ac.id

**Data Availability Statement:** All relevant data are within the paper and its Supporting Information files.

## Abstract

Tomato commodity ranks fifth regarding vegetable export volume and value in Indonesia. The main issues with tomato production in Indonesia are seemingly its lack of variations, quantity, quality, and supply continuity. This study aimed to analyze the comparative and the competitive advantages of tomato farming, evaluate the government policy on inputs, outputs, and input-output sectors, and formulate strategic for transforming the comparative advantage into a competitive advantage. Policy Analysis Matrix (PAM) and sensitivity analysis were employed as the method with the 75 farmers as the respondents and their Focus Group Discussion (FGD) involving farmer groups, agricultural extension workers, traders, and the Agricultural Service Offices in the respective regencies. The results showed that tomato farming has both its comparative and competitive advantages. Its comparative advantage was higher than its competitive advantage in both dry and wet seasons. In general, tomato farming's comparative and competitive advantages outside Java were higher than those in Java. Meanwhile, the divergence effects of tomato agribusiness were more beneficial to consumers than producers. It indicates that improving domestic tomato production was more profitable for Indonesia than importing this commodity in terms of domestic resource use. The sensitivity analysis shows that tomato farming was relatively stable regarding productivity and price changes. The suggested strategic policies to apply are transforming tomato farming's comparative advantage into competitive advantage through productivity enhancement, improvement of distribution efficiency, less market distortion, and government incentives.

**Funding:** The author(s) received no specific funding for this work.

**Competing interests:** The authors have declared that no competing interests exist.

## 1. Introduction

Globalization and trade liberalization are increasingly providing new opportunities for farmers and business actors in the supply chain of horticultural commodities. Still, at the same time, they also create several new challenges. In the era of free trade, one of the apparent challenges is a tight and fierce competition that requires each supply chain player to possibly innovate and contribute to horticulture product improvement. It is also known that trade liberalization provides opportunities in terms of market expansion for an export product due to reduced trade barriers by each country [1]. On the other hand, the existence of trade liberalization can also be a significant constraint problem for some countries that cannot produce horticultural products competitively. Those countries will consequently become a market for horticultural products from highly competitive countries [2]. Therefore, having product competitiveness is considerably needed now.

The determinants of product competitiveness are caused by the performance of the entire product value chain at the domestic and global levels [3, 4]. Previous research has demonstrated that a product's competitiveness depends on how well the complete value chain performs, from the early stage to the final stage, both nationally and internationally [5, 6]. At the domestic level, product competitiveness is measured by one's capacity to add value. The creation of value-added products can be sought by improving technical skills and business governance capabilities, strengthening institutional capacity, entrepreneurial abilities, and active involvement of stakeholders in the industry. Globally, the value chain's input-output composition, governance, geographical coverage, and the management structure of industrial organizations determine competitiveness at the global level. Similarly, in some value chains of agricultural products, the analysis and the direction of progress are increasingly oriented toward global markets. Therefore, the value chain is often referred to as the global value chain (GVC) [5–7]. Meanwhile, based on Gultom, 2020 [8], the added value of Indonesia's agricultural and food products could be optimized through involvement in regional and global value chains such as the Regional Comprehensive Economic Partnership (RCEP). Fresh horticultural products and their derivatives have opportunities to contribute and participate in regional and international value chains because of their relatively high indices of forward and backward linkages in the aggregate [9, 10].

Fresh tomato commodity is among the most traded horticulture commodities. It has been the most dynamic value chain of horticultural exports from Mexico to the USA market [11]. Besides Mexico, Jordan is the 4th largest exporter of fresh tomatoes globally [12]. In Jordan the highest vegetable production is tomato production, with a share of 43.4% of the total vegetable production. A study conducted by Gibba, 2017 [13] revealed that the Netherlands, Spain, and Mexico were the main exporters of vegetables, along with 54% of all exported products; fresh tomatoes dominate with more than 40% of the value of the vegetable trade there. Several empirical studies related to the competitiveness of exports to certain countries' markets focused on competitiveness from the supply side, where it only analyzed groups of competing countries [14–16]. Capobianco-Uriarte et al., 2021 [17] conducted research on the competitiveness of tomato exports from the market demand perspective by analyzing the main customer market in the context of the European Union using the Constant Market Share (CMS) methodology to analyze variations in exports to assess the competitiveness of the market. The study results concluded that Spain and Belgium can compete for tomato commodities in the European market. Steel, 2013 [18] looking at both the competitiveness of the supply side and the demand side of the business strategy with the objective of return on investment (ROI), concluded that the function of ROI simultaneously represents the investment allocation theory and product design specifications. Competitiveness analysis by looking at comparative and

competitive advantages using PAM is a competitive advantage from the supply side [19–22]. The advantage of analyzing competitiveness using PAM is able to observe profit analysis from a private (financial) perspective and, at the same time, from a social (economic) perspective, the competitive advantage as well as the comparative advantage, and see the impact of good divergence effects caused by government policies as well as market failure.

Competitive analysis of horticultural commodities can be examined from two directions, particularly the financial perspective (private) and the economic perspective (social) [23–25]. The results of a study carried out by Bayzidnejad et al. (2021) [26] mentioned that the indicators of competitiveness from a social perspective showed that several agricultural commodities in Urmia had a comparative advantage in wheat, barley, sunflower, tomato, beans, and kidney beans production. In terms of the net social profit (NSP) index, domestic resource cost ratio (DRCR), and social cost-benefit (SCB) index, the tomato production had a higher comparative advantage than those of red beans, peas, sunflower, buckwheat, and barley. The results of another study on the competitiveness of some vegetable commodities revealed that shallot, potato, and chili farming in production center areas in Indonesia were relatively good from the perspectives of competitive and comparative advantages [23–25]. In general, the competitive advantage of vegetable commodities including tomato is lower than its comparative advantage, which causes disincentives for farmers to increase tomato production.

Although tomato is not a strategic national horticulture commodity, it is an export promotion commodity used as a raw material for the processing industry and seasoning in the instant noodle industry. The nutritional composition of a tomato of 100 grams comprises of a protein (1 g), carbohydrates (4.2 g), fat (0.3 g), calcium (5 mg), phosphorus (27 mg), and iron (0.5 mg), vitamin A (carotene) 1500 SI, vitamin B (thiamine) 60 g, vitamin C 40 mg [27]. According to Porter (1990) [28], sophisticated and demanding buyers, namely smart and critical consumers, will provide valuable information to producers about the quality of products and services consumers need. Based on these references, today's global consumer demand requires a variety of more complete and detailed product attributes such as nutritional, safety, value, ecolabel, product traceability, and humanistic. In addition, tomato is a commodity with broadbackward and forward linkages. A number of key issues in tomato production are intensive fertilizer application, pesticides, and labour. Meanwhile its marketing issues include relatively low prices, higher shipping expenditures, perishable, and long distances between the farmer's land and the markets.

Reasons to conduct the study on tomato comparative and competitive advantages analysis are (a) tomato commodity ranks fifth of Indonesia's exported vegetables values and volumes, and its export markets are Kuwait, Singapore, Malaysia, and Timor Leste [29]; and (b) based on its comparative and competitive advantages, it can possibly help the government to formulate the policies for improving this commodity using the domestic resources efficiently such that it can also become promising export commodity. The novelty of this study is to gain tomato production competitiveness status based on its comparative and competitive advantages with its distribution in Java and outside Java areas as well as the growing seasons. It also reveals a sensitivity analysis of productivity and tomato price variables. The other novelty is to put forward the strategy to change its comparative advantage into a competitive advantage, which is useful for efficiently expanding tomato production based on domestic resources.

The following are a number of considerations to take when measuring tomatoes' competitive and comparative advantages, as well as the impacts of divergence on the tomato commodity system: (1) Whether there is a comparative advantage for tomato growing given the inherent potential of available resources and technology. (2) Whether tomato cultivation has competitive advantage in the market manufacturing hubs have the most comparative and competitive advantages during the dry and rainy seasons. (3) What effects government actions

have on input, output, and the overall dynamics of input and output. (4) What strategic initiatives can be put into place to turn tomato farming's comparative advantage into a competitive advantage.

This research paper endeavours to achieve the following objectives: (1) to conduct a comprehensive analysis of the financial and economic viability of tomato farming; (2) to evaluate the competitive and comparative advantages inherent in tomato farming; (3) to assess the impacts of policies and divergence on the input, output and input, and output sectors simultaneously; and (4) to formulate strategic policies aimed at transforming the comparative advantage of tomato farming into a sustainable competitive advantage within the market.

## 2. Literature review

Several economists continue to explore the theory of competitiveness such as disclosed by Esterhuizen, 2006 [30] including Staffan Linder, 1961 [31] with "Overlapping Demand"; Raymond Vernon (1966) [32] with "Product Cycle Theory"; Krugman (1979) and Lancaster (1979) [33, 34] who produced the "Economic of Scale" theory; and Michael Porter (1990) [28] with "Competitiveness Theory" which states that welfare is created through choices. Among business experts, it is argued that the concept of competitiveness or competitive advantage doesn't only a theory of economics though additionally encompasses business ideas. Teece, 2009 [35] suggests that the importance of integrating strategy analysis and business model is to maintain the advantage gained through the creation of innovative business models.

Competitive advantage is the capability to distribute goods and services at the time, place, and form intended by purchasers either in domestic or international markets at prices that are good or better than the other suppliers while at the same time getting the lowest opportunity cost on the employed resources [36]. Furthermore, Esterhuizen et al., 2006 [37] define competitiveness as the competence of an industrial sector to contest effectively to reach sustainable progress in the international setting and simultaneously achieve the lowest while earning the minimum opportunity cost of return on utilized capital. The results of a study by Esterhuizen & van Rooyen (2006) [38] on the factors that affect the competitiveness of the wine industry in South Africa show that the factors that have a positive impact on competitiveness are the production of high-quality products, strict regulatory standards in the industry, industry efficient support, and the availability of competitive local primary input suppliers. Meanwhile, the main factors that harm the competitive success of this industry are fluctuations in exchange rates, confidence in the political support system, the competence of personnels in the public sector, crimes and labour policies, and the growth and size of the local market.

Some people view competitiveness as a business concept used as the basis for many strategic analyses to improve company performance [39]. Based on the micro-innovation perspective [40], a model called the "dynamic competitive advantage formation mechanism model" focuses on the fact that in a competitive market, firms could even maintain a long-term competitive edge by dominating limited resources that can only be replicated or modified. However, firms must strengthen their segments and sub-abilities as well as enlarge the impact of micro-innovation across digital technologies, thus achieving and sustaining a competitive advantage across continuous micro-innovation. Michael Porter, Harvard University Professor of Economics and Strategic Management Expert [28], we treat Porter as a "general" scholar in his publication entitled "*The Competitive Advantage of Nations.*"

The competitive analysis gives useful information on a commodity's competitiveness against others in global markets [19, 21, 22]. The results of the analysis of the competitiveness of a commodity can also be identified as to whether government policies on the commodity system are protective (incentives) or detrimental (disincentives) to farmers as producers

[19, 21, 22]. Tupkanloo's (2015) [41] findings convincingly showed that Khorasan Razavi Province, Iran, had a comparative advantage in producing tomatoes and cucumbers with a Domestic Resource Cost Ratio (DRCR) value of less than one. The policy of the government indicators in the field of inputs for tomato and cucumber farming created the results of the Nominal Protection Coefficient on Input (NPCI) > 1, indicating the tax burden was indirectly charged to producers. Effective Protection Coefficient (EPC) <1 reflects that government intervention was detrimental to tomato and cucumber-producing farmers. Research results from Saptana et al. [23, 24] showed that the overall effect of government decisions or market forces in the input and output fields and simultaneous input and output on shallot and potato farming was unfavourable to farmers and advantageous to consumers. Meanwhile, the results of the study of the impact of government policies as a whole on the shallot commodity system [23] and the chili commodity system in production centres in Indonesia [25] provided contra- dictory results, namely that the government policies were more helpful to farmers.

Thus, the results of the analysis of the competitiveness of a commodity system also provide an evaluation of the government policies' performances in agricultural development, whether to protect producers, consumers, or the balance between the two [19, 21–23, 25, 26]. Further- more, it can be said that the results of the competitiveness analysis using PAM are also useful for determining the types of policy options that can be implemented, such as input transfer policies, output transfers, input subsidies, output subsidies, and effective protection policies for a commodity system.

Indicators used to determine competitive advantage are those of private profitability and Private Cost Ratio (PCR), while to determine comparative advantage indicators used are those of social profitability and Domestic Resource Cost Ratio (DRCR). The computation involved in this analysis is inherently intricate and necessitates meticulousness due to its incorporation of private and social prices, private and social costs, and private and social revenues. Moreover, it entails the careful allocation of input components into both tradable and domestic factors. PCR is the ratio of domestic input costs at private prices to the difference between tradable input revenues and expenses at private prices [19, 22]. PCR, as a metric of competitive advan- tage, provides a quantitative measure of the extent to which domestic resources can be con- served in order to produce a single unit of output at private prices.

The results of a study conducted by Chowdhury (2020) [42] on the improvement of PAM for Bangladesh's fruits and vegetables export process observed that Bangladesh had high pros- pects and potential for vegetable exports due to a high demand from international foreign markets. Some of the obstacles faced were high transaction costs, low product quality, and the imposition of various Sanitary and Phytosanitary criteria by importing countries. The results of this study follow those carried out by Saptana et al., 2021; Saptana et al., 2022a; Saptana et al., 2022b [23–25] that shallots, potatoes, and chillies in production centres in Indonesia had moderate to highly competitive and comparative advantages with PCR and DRCR coefficients less than one. In general, their competitive advantages are lower than their comparative advantages.

According to the results of PAM, which evaluates the comparative advantage of production and gauges the competitiveness of important crops in Urmia County, Iran, this area had a comparative advantage in growing irrigated wheat, rainfed wheat, rainfed barley, sunflower, tomato, chickpeas, and kidney beans. It is regardless to the fact that there are two of the princi- pal crops in the area, irrigated barley and sugar beet cultivation in the county which lacked a comparative edge. In this study, three comparative advantage indicators were used, i.e., NSP, DRCR, and SCB. Tomato production had a larger comparative advantage than other food commodities, according to the NSP index [26].

Roeun (2018) [43], using a combination of value chain analysis (VCA) and a policy analysis matrix (PAM), found that tomato producers in Cambodia confronted a high level of transaction costs as a result of poor coordination in the domestic tomato value chain. Based on the PAM approach, the gains in private pricing were significantly lower than those in social prices, with transaction costs decreasing private earnings by roughly 60 percent. The production of tomatoes in Cambodia provides a competitive advantage over imports, as determined by DRCR estimates, which reveal a DRCR coefficient below 1. Relevant to that study, another study of Rinaldi and Nugrahapsari, 2017 [44], showed that vegetable growing in Bali retained comparative and competitive advantages, particularly in farming commodities such as red chili, cayenne pepper, and tomato, with PCR and DRCR coefficient values of less than 1. Moreover, the results of a study conducted by Widiastuti and Umar, 2021 [45] on the impact of combining biopesticides and fertilizers provided interesting results showing that the treatment gave the highest tomato productivity. It indicated that industrial biopesticides could be a good alternative to replace chemical pesticides, while dependence on inorganic fertilizers was difficult to avoid. Minakov and Nikitin, 2019 [46] revealed that areas of increasing agricultural production of high economic value commodities such as fruits, vegetables, milk, and their derivatives could be concentrated on increasing productivity, increasing agricultural organizational capacity, transferring from industrial agriculture to innovative development, modernizing production facilities, build modern facilities, the efficiency of commercial activities and conducive government policies.

The tomato value chain in Egypt was divided with a tiered system and poor linkages because tomato producers' access to the final market inhibited their capacity to obtain a fair share of the price paid by consumers [47]. Meanwhile, the main obstacles to increasing production and marketing faced by tomato farmers in Ghana's Upper East Region's Talensi Nabdam District were a lack of reliable markets and of access to credit sources, which made it hard for tomato farmers to scale up, increase a production capacity through the adoption of advanced technology and incapable of anticipating, mitigating and managing farm risks [48].

The global effect of COVID-19 pandemic showed that limited awareness of achievement in the consumers' value chain gained an impact on a firm [49]. The study's findings on operational efficiency, system performance, and profitability were all important. Attaining long-term profitability, robust growth, and efficient solvency require further thinking about enhancing and managing conflicts between value chain systems. As a result, the corporation must augment operational efficiency while placing profitability on its priority list [50]. Profitability outcomes served as valuable indicators of good business growth in global production networks. However, profitability measurements only showed results for a certain period and did not indicate the company's long-term performance [51].

The results of a tomato competitiveness study by Dorogi, 2021 [52] showed that Hungary did not have a comparative advantage in terms of tomato products on the global market with an RCA index in the range of $0 < RCA \leq 1$, while countries that had a stable comparative advantage are Spain and the Netherlands with an index RCA exceeds one but does not reach 2. Similarly, Familusi et al., 2015 [53] conducting a comparative analysis of the comparative advantages of tomato production between Mozambique and South Africa with the application of DRCR obtained the result that Mozambique has a comparative advantage in producing tomatoes compared to South Africa. A study by Ali et al., 2020 [54] concluded, based on the results of the PAM analysis on the Tomato and Potato commodity system in Egypt during the average period (2011–2017), both have comparative and competitive advantages, each with a DRCR coefficient < 1 and PCR < 1.

## 3. Methodology

### 3.1 Research framework

Ricardian comparative advantage is one of the primary foundations of international trade theory [55]. Such concept of competitiveness is based on the idea of comparative advantage introduced by Ricardo in 1823, later known as the Ricardo Model or *"The Law of Comparative Advantage."* According to him, even if one country is less efficient than another in producing both goods, there is still a basis for trade that benefits both sides. The first country must concentrate on producing and exporting commodities with a lower absolute loss or having a comparative advantage and importing commodities with a higher absolute loss or having a comparative disadvantage [37, 56].

Haberler intensified Ricardo's theory of comparative advantage by introducing the concept of comparative advantage based on the value theory based on labour costs [55]. According to this hypothesis, products are exchanged for each other based on the relative quantity of labour included in them. Haberler identifies Ricardo's essential thesis: that every country may produce commodities based on its natural resources, labour, and capital factors, produce more than it needs, and exchange its excess with other countries for goods it is unable or unwilling to produce. In any case, the difference in opinions rests in the assumptions utilized by Ricardo to show his theory, especially the theory of labour costs.

The Heckscher-Ohlin hypothesis, an economic theory of comparative advantage in international commerce, postulates that nations endowed with a lot of capital and relatively little labour are more likely to export capital-intensive goods while importing labour-intensive ones. In contrast, nations with a lack of capital and a surplus of labour typically buy products with high capital requirements while exporting goods with high labour demands [57, 58]. International trade through export and import activities is an activity of buying and selling goods and services between countries due to the abundance of different resources between countries [59].

Since the beginning of time, competitive advantage has been viewed by economists as the result of a convergence between market imperfections and comparative advantage [19–22, 60–62]. Comparative advantage is a metric of prospective competitiveness or advantage when it comes to competitiveness which is reached if the economy is free of any distortions. Commodities that have a comparative advantage are also said to have an economic efficiency. Competitive advantage is a measure of a company's ability to compete in current economic conditions. Comparative advantage is inextricably tied to the concept of economic (social) feasibility, whereas competitive advantage pertains to a corporation's private financial feasibility [22, 60–62]. Sources of distortion that can interfere with the level of competitiveness include (1) governance policies, both direct (tariffs) and indirect (such as regulations); and (2) market distortion, due to market imperfections, for example, the existence of a monopoly or monopsony as well as an oligopoly or oligopsony in the market. The establishment of food safety standards policies by rich countries might have a double impact on developing countries. On the one hand, it may impose trade restrictions; on the other hand, it may serve as a motivator for improving food safety management capabilities and building a competitive position in the high-value product market [63].

The Policy Analysis Matrix (PAM) can be used to examine the influence of policies on the agricultural commodities system as well as the measurement of competitiveness from the standpoint of competitive and comparative advantage [19, 21, 22, 62]. The results of PAM analysis can generate several results indicators of competitiveness, such as private profitability (PP), social profitability (SP), *Private Cost Ratio* (PCR), and *Domestic Resource Cost Ratio* (DRCR). PP and PCR are indicators of competitive advantage that demonstrate the system's

ability to pay for domestic resource costs and remain competitive at private prices, while SP and DRCR are measures of comparative advantage and also the role of public policy on a commodity's farming system [19, 21, 22, 62].

In the global marketplace, the degree of competitiveness observed in a country, industry, or product is formed by two fundamental factors: comparative advantage and competitive advantage [62]. The abundance of natural resources is a source of comparative advantage, whereas production capacity or strategic advantages are sources of competitive advantage. The Policy Analysis Matrix (PAM) can be used to efficiently examine and assess comparative advantage, competitive advantage, and the effects of government interventions on the Indonesian commodity system [19–21]. This approach has been widely used in assessing comparative and competitive advantages and the impact of government policies on the performance of agricultural commodities farming [23–25, 62, 64–66].

According to Monke and Pearson, 1995; Pearson et al., 2005 [20, 21], three main discussions can be explained through the PAM approach related to the agricultural commodity system. First, PAM can be used to measure the impact of government policies on the level of competition at various levels of profit, financial, and economic efficiency, the impact of comparative advantage on investment strategy, and the impact of technical progress on agricultural growth. Secondly, PAM can be used to measure economic or comparative advantage and financial or competitive advantage in agricultural investment based on the suitability or technological advantage and the condition of natural resources, including agro-climate. Based on these advantages, the policy on natural resource use is feasible or not to be developed through domestic or foreign investment. The attractiveness of investment can increase efficiency and accelerate national income growth. Thirdly, PAM is closely related to a series of problems in the allocation of research funds or research in agriculture. Using PAM, a researcher can determine what strategic policies can be applied to improve agricultural production, reduce social costs, and expand farmer welfare.

## 3.2 Research locations

The fundamental goal of this research article is to thoroughly examine competitive advantage, comparative advantage, and the effects of policies on input and output dynamics. The implementation of this research is focused on the location of the tomato commodity area development program, which is driven by the Directorate General of Horticulture, Ministry of Agriculture of the Republic of Indonesia. The location study selected 6 study locations. There were 3 (three) regencies in Java, namely: Bandung Regency, representing the production centre area of West Java; Banjarnegara Regency as the production centre area of Central Java Province; Kediri Regency as the production centre area in East Java; and three (3) regencies outside Java, namely Tanah Karo Regency representing production centres in North Sumatra, Tabalong Regency representing production centres in developing South Kalimantan Province, Pinrang Regency representing production centres in South Sulawesi.

BPS data show that tomato production in Java Island provides a production share of 464,865.4 tons (41.71%), Sumatra Island 392,964.2 tons (35.26%), Kalimantan Island with a production share of 27,789 tons (2.49%), and South Sulawesi with a production share of 156,608.5 tons (14.07%) of Indonesia's total production of 1,114,399.5 tons (100%) [67]. In addition, this site selection also considers that these areas have exported tomatoes or have the potential to export tomatoes. Therefore, the selected research location could considerably represent the demonstration of tomato production in Indonesia. The study was conducted from April to December 2020, i.e., during the rainy season (MH) 2019/2020 and the dry season (MK) 2020.

### 3.3 Sampling method

This study uses a mathematical approach, i.e., PAM, instead of a statistical approach so that it does not require a certain number of respondents (N) but rather emphasizes the strength of respondent representation. The sampling methods used in this research are purposive sampling for farmer respondents and the snowball method for trader and exporter respondents. In this case, respondents were purposively selected to representing the physical conditions and socioeconomic characteristics of each study location. The study employed a PAM technique, integrating targeted survey-based FGDs with tomato farmer group management as well as interviews with tomato farmer households. The number of samples was 15 farmer households for each regency on Java Island, and as many as 10 farmer households for each regency in areas outside Java, or overall 75 respondents altogether. The sample selection of farmer households represents large-land farmers, medium-land farmers, and smallholder farmers. From the farmers in each class, using relatively homogeneous technology, and also the agricultural extension workers and the local regency agricultural officers, the data and information were obtained and gathered.

In addition, a Focus Group Discussion (FGD) was also carried out at each research location, consisting of heads and administrators of farmer groups, leaders of advanced farmers and farmers, field agricultural extension workers, regency agricultural office officers, village collector traders, inter-regional traders, wholesalers and exporters. Each respondent surveyed, namely the administrators of the federation of farmer groups, the administrators of farmer groups, the farmers, and the traders in each research location, was interviewed to describe the existing conditions in the field. In doing so, The research team was accompanied by agricultural officers and local agricultural extension workers. The respondents surveyed were also briefed about the aims and objectives of research activities, as well as the benefits of research activities in agricultural development policies and also the importance to increase the competitiveness of tomato farming.

The method used to obtain respondents' consent was done orally and witnessed by local agricultural officers and agricultural extension workers. The confidentiality of all the data is guaranteed, and the data used are only those related to the purpose of the study. This study not only uses the FGD method but also uses limited surveys and snowball sampling methods from the farmer to exporter level, with the following details: (i) for farmer respondents there were a total of 15 respondents by regency in Java Island and 10 respondents per regency for areas outside Java Island; (ii) In this study, a limited survey was also conducted through interviews with traders at various levels, 2 collecting traders, 2 interregional traders and 2 wholesalers, 2 retailers, 1 exporter per each location; and (iii) In addition, an FGD was conducted consisting of 10 business actors in each regency where the study was conducted, which included: heads of farmer groups and administrators of farmer groups, extension workers, coordinators of field extension workers, officers of the Agriculture Office in each regency, traders, packing house entrepreneurs, and exporters.

The tomato farming, comparative advantage, and competitive advantage analysis or examination were done on a per-hectare basis and with considering both the dry and rainy seasons. The Focus Group Discussion (FGD) sampling method involved the federation management of the farmer group, the management of the farmer group itself, and the farmer group members to capture information on farming performance as well as input and output market structure for determining private prices and social prices, both for traded inputs and domestic factors. In addition, the tomato traders, consisting of collecting traders at the village level, inter-regional traders, wholesalers, and retailers, were interviewed to collect information on the distribution chain and marketing margins for calculating private and social prices. The survey of

tomato commodity traders was carried out using the snowballing method from the collector to the retailer levels.

## 3.4 Analysis method

This study examined comparative advantage, competitive advantage, and the impacts of government policies on tomato farming systems in Indonesia. The analytical method used was the Policy Analysis Matrix (PAM). This approach assessed both the financial perspective (private perspective) and the economic perspective (social perspective) [19–21, 23–25, 62]. The PAM method produces 13 indicators, as follows: Private Profitability (PP), Social Profitability (SP), Private Cost Ratio (PCR), Domestic Resource Cost Ratio (DRCR), Output Transfer (OT), Nominal Protection Coefficient on Tradable Output (NPCO), Input Transfer (IT), Nominal Protection Coefficient on Tradable Input (NPCI), Factor Transfer (FT), Effective Protection Coefficient (EPC), Net Transfer (NT), Profitability Coefficients (PC), and Subsidy Ratio to Producer (SRP).

The OT and NPCO indicator values indicate the effects of output transfer on the tomato farming system in Indonesian production centres. OT describes the transfer of output that arises through the difference between the output value at private prices and the output value at the social price received by farmers as producers or paid by consumers. The large negative OT value reflects the transfer of output from farmers as producers to consumers. From the output side, farmers as producers are relatively disadvantaged compared to the more benevolent consumer society. NPCO is calculated by dividing revenues at private prices by revenues at social prices. The net present value coefficient of output (NPCO) can be used to evaluate the effect of public policy incentives that produce production value disparities as assessed by private prices and social prices. Government initiatives in the output sector can take the form of trade policies such as customs, value-added tax (VAT), import tariffs, subsidies, and direct financial support. OT is the difference between acceptance at private prices and acceptance at social prices. NPCO is an output transfer indicator determined by dividing revenues calculated using private prices by revenues calculated using social prices.

OT and NPCO for tomato farming in Indonesian production centres were calculated using the PAM measurement matrix analysis results, as depicted in Table 5. The OT was negative in both rainy and dry seasons, and the NPCO was lower than one. Tomato farmers obtained a real output price less than it ought to be when compared to fully competitive market circumstances or when there was no distortion. Distortions could be caused by either public policy interference or market distortions. In other words, farmers deal with disincentives in tomato production on the output side. The market system for vegetable goods, including tomatoes in Indonesia, was oligopsony. It was shown by the existence of a relatively large number of farmers or vendors but few purchasers [23–25, 68].

The PAM calculation stage consists of five steps [23–25]. First, compiling the physical input-output structure of tomato farming, including labour use, equipment depreciation, land rent, and capital interest. Second, estimating the shadow price (social price) of the input and output of tomato farming, including labour, equipment depreciation, land rent, and capital interest. Third, separating all costs of tomato farming into tradable inputs and domestic factors; Fourth, calculating the cost and revenue of tomato farming financially, privately, economically or socially; and Fifth, computing and analyzing the indicators that can be generated from PAM, both indicators of profit, comparative and competitive advantage, the impact of policies in the input sector, the impact of policies in the output sector, and the impact of policies on input and output simultaneously on the tomato commodity system.

If the PCR value is less than one, the tomato business activity is said to have a competitive advantage; the smaller the PCR coefficient, the more efficient it is. DRCR is calculated as the proportion of domestic input costs to social prices minus the difference between tradable input income and social costs [19, 22]. DRCR is a measure of comparative advantage that describes how much domestic resources can be saved to produce one unit of foreign exchange. Without government intervention or market conditions in perfect competition, tomato farming activities are said to have a comparative advantage if the DRCR value is less than 1.

The sensitivity analysis was necessary to cope with the limitation of the mathematical approach in PAM. Thus, it analyzed the changes in productivity or in prices that cause tomato farming in each location not to get a competitive advantage (PCR = 1) or not to get a comparative advantage (DRCR = 1). This analysis was relevantly needed to observe the stability of both the competitive and the comparative advantages of tomato farming in each of the locations of the study, when there are changes in the productivity level and the tomato selling price.

**3.4.1 Components of tradable input and domestic factor allocation.** Tomato farming production costs are divided into tradable input costs and the costs of domestic factors, including production input costs, labour, equipment depreciation, land rent, and capital interest. The first category of expenses includes marketable inputs that can be exchanged on the international market, whereas the second category includes domestic components that are not traded on the worldwide market.

The following attributes can identify the marketable goods: (a) commodities that are currently traded in international markets, either exported or imported; (b) commodities that serve as substitutes for other types of products traded in international markets; and (c) commodities that receive governmental protection measures [20, 21, 69, 70].

Theoretically, according to Monke and Pearson (1989, 1995); Pearson et al., 2005 [19–21] and has been widely applied empirically [24, 25, 71–73], there are two techniques of distributing tomato cultivation expenses into tradable input costs and domestic variables, namely the overall approach and the direct approach. The total method assumes that each tradable production input cost consists of domestic and tradable input components. This method is used to assess the effect of a policy on the performance of the commodity system. The direct method entails assessing all tradable input costs, both imported and domestically generated, as a collection of tradable input costs. This strategy is adopted if extra marketable inputs, including imports and domestic production, can be fulfilled through international commerce. It is straightforwardly done in this paper. This technique was adopted for a variety of reasons, including rising trade liberalization, and increasing demand for tradable inputs that may be supplied from global markets.

A direct method was applied in the PAM study of the tomato agricultural system at the research location. This direct approach has been widely applied [23–25, 62]. Thus, fresh tomato output was considered to be 100 percent marketable, but tomato seeds, Urea/Za, SP-36/TSP, Potassium Chloride, NPK/PONSKA, insecticides, and plastic mulch were expected to be 100 percent tradable. Meanwhile, solid organic fertilizer, liquid organic fertilizer, dolomite, raffia rope, stakes, labour, land rent, and capital interest were considered to equal to 100% of the cost of the domestic component.

The distribution of tradable input costs and domestic factor costs for transportation operations was based on the findings of a field survey conducted through interviews with business administration actors at various levels. Transportation expenses reflecting the rental value of transportation equipment were included as components of tradable inputs, whereas labour costs in the process of carrying commodities were included in the domestic factor cost. The allocation of tradable input components and domestic variables for post-harvest handling costs was based on data mining results from direct interviews with farmers and participants in

the tomato commodity trade system. Material costs were allocated as trade inputs, whereas labor costs were assigned as domestic components. The land rent and capital interest were fully included as foreign components. Appendix 1 in S1 Appendix shows the outcomes of allocating tomato cultivation cost components into tradable inputs and domestic variables.

**3.4.2 Determination of social pricing.** To assess a tomato production system's private and social viability, private and social pricing must be determined. As a result, two prices are established for each input and product of the tomato farming system: private pricing and communal pricing. The private price is the amount that producers receive from the market when they sell their goods and/or the amount that producers pay when they purchase the necessary production inputs. On the other hand, the social price is the price established when the market mechanism performs ideally or when the economy reaches a condition of general equilibrium characterized by full employment [19–21, 69, 70]. It is rare to establish empirically fully competitive market conditions in which the counter-cost matches the market price. Thus, opportunity costs must be taken into account, indicating that social prices must be modified to account for the impact of governmental interventions and market inefficiencies. The shadow price is established by eliminating any inefficiencies in the tomato commodity system, including distortions brought on by government actions (subsidies, import tariffs, export tariffs, and value-added taxes). It is possible to approach the tomato commodity in this study, where it is a traded good with a border price. For example, f.o.b. pricing is utilized for export items. Prices for exported and imported commodities are expressed in terms of cost insurance freight (c.i.f.) and "free on board," respectively, with different modifications made to reflect the degree of competition between them. Meanwhile, the investigation focuses on assessing the opportunity cost or the average price associated with domestic components in each sampled research area. The calculation method for each input and output shadow price is described in Table 1.

Entirely and in detail, the results of assessing the social prices of inputs and outputs in the tomato farming system are presented in Appendices 2 and 3 in S1 Appendix.

**3.4.3 PAM compilation.** The following five sequential processes make up the computation method for the Policy Analysis Matrix (PAM) in tomato farming: (1) estimating the total physical input and output of the under-study agricultural system; (2) classifying expenditures into domestic and tradable input components; (3) calculating income levels; (4) assessing the social prices of inputs and outputs; and (5) performing calculations and analysis of the 13 indicators derived from the PAM analysis, as shown in the analysis table (Appendices 4 to 15 in S1 Appendix).

The PAM was created following the collection of all farmer-level data and actors from the tomato trade system. The PAM matrix was created by combining the physical input-output framework at the level of the farmer, budgetary expenses, private and public income, as well as information on transportation costs collected from trading system members. This estimate productivities both private and societal advantages. The effect of governmental policies on the input and output variables, including inputs and outputs of the tomato commodity system, is investigated.

The outcomes of the PAM analysis provide insights into profitability for both individuals and society, comparative and competitive advantages, as well as the effects of governmental policies on the input and output facets of the tomato commodities system. Table 2 shows the PAM table at each site.

## 4. Results and discussion

Some economists view competitive advantage as a result of comparative advantage and market distortion caused by market errors and government policies [19–21]. The concept of

**Table 1. Approaches of calculation method for input and output shadow price of tomato commodity.**

| No. | Input or output description | Social price approach |
|---|---|---|
| 1. | Tomato seed | Average of actual tomato seed price in research locations. |
| 2. | Solid and liquid organic | Average of actual solid and liquid organic fertilizer prices in research location |
| 3. | Urea | f.o.b. price in 2020, i.e., US$ 0.243/kg, converted to IDR as Rp 3,540/kg |
| 4. | SP-36 | c.i.f. price in 2020, i.e., US$ 0.242/kg, converted to IDR as Rp 3,522/kg |
| 5. | Potassium Chloride | c.i.f. price in 2020, i.e., US$ 0.307/kg, converted to IDR as Rp 4,476/kg |
| 6. | NPK | c.i.f. price in 2020, i.e., US$ 0.391/kg, converted to IDR as Rp 5,698/kg |
| 7. | Dolomite | Average of dolomite price in research location because it was not traded internationally. |
| 8. | Pesticide | Average of actual pesticide price in research location. |
| 9. | Plastic mulch | Average of actual plastic mulch price in research location. |
| 10. | Stake | Average of actual stake price in research location |
| 11. | Raffia rope | Average of actual raffia rope price in research location |
| 12. | Farm workers | Average of actual wage value in research location |
| 13. | The value of land rent | Average of actual farming rent in research location |
| 14. | Irrigation cost | Average of actual watering cost in research location |
| 15. | Interest rate | The People's Business Credit Program has a fixed interest rate. When we evaluate this interest rate and how prices are rising (inflation), we see that the total interest rate is 2.59% every 5 months. This means that when people borrow money from the tomato growing program, they only have to pay back the 1.89% interest for each tomato growing season (which lasts 5 months). |
| 16. | Rupiah exchange rate | Based on the average rupiah exchange rate in 2020, i.e., Rp 14,577/US$. |
| 17. | Output price of tomatoes | A value of Rp 22,451/kg was obtained by taking into account the cost, insurance, and freight (c.i.f.) price of U$ 0.154/kg and then converting it using the dollar-to-rupiah exchange rate. |

**Table 2. Policy analysis matrix (PAM).**

| Description | Revenue (IDR/ha) | Cost (IDR/ha) | | Profit (IDR/ha) |
|---|---|---|---|---|
| | | *Tradable input cost* | *Domestic factor cost* | |
| Private Price | A | B | C | D |
| Social Pricing | E | F | G | H |
| Impact of Divergence and Policy | I | J | K | L |

Notes

1. Private Profitability (PP): D = A–(B + C)

2. *Social Profitability* (SP): H = E–(F + G)

3. *Private Cost Ratio*: PCR = C/ (A–B)

4. *Domestic Resource Cost Ratio*: DRCR = G / (E–F)

5. *Output Transfer*: OT = A–E

6. *Nominal Protection Coefficient on Tradable Output*: NPCO = A/E

7. *Input Transfer*: IT = B–F

8. *Nominal Protection Coefficient on Tradable Input*: NPCI = B / F

9. *Factor Transfer*: FT = C–G

10. *Effective Protection Coefficient*: EPC = (A–B) / (E–F)

11. *Net Transfer*: NT = D–H

12. *Profitability Coefficient*: PC = D / H

13. *Subsidy Ratio to Producer*: SRP = L/E.

competitive advantage is financial feasibility (private), while comparative advantage is related to the economic (social) feasibility of an economic activity [60–62]. In principle, competitive advantage and comparative advantage are complementary in determining competitive performance and maintaining sustainability in global trade [74]. The competitive status of the tomato commodity farming system is beneficial in determining the direction of development policies on productivity and efficiency enhancement, as well as appropriate government policies.

## 4.1 Private and social profitability

The analysis results of costs and financial benefits (private) showed that tomato farming profit in the production centres in Indonesia decreased in both the rainy season (2019/2020) and the dry season (2020). In general, economic benefits (social) were higher than financial benefits (private), and profits during the rainy season were lower than those during the dry season. It was due to its being very susceptible to high rainfall, especially during flowering and fruiting. Tomato production decreases during the rainy season due to flowering and pest and disease attacks and an extreme climate change. The intensity of pest and disease attacks is knowingly higher during the rainy season. The commonly found types of pests, especially during the rainy season, are Ground caterpillars (*Agrotis ipsilon* Hufn.), Tomato fruit caterpillars (*Helicoverpa armigera* Hubn.), Whiteflies (*Bemisia tabaci* Genn.), Soldier caterpillars (*Spodoptera litura* F.), and Leafminer Caterpillar (*Liriomyza huidobrensis* Blanchard) [75]. Meanwhile, the types of tomato plant diseases consist of: Slinging Down Disease caused by fungal pathogens (*Pythium sp.*, *Rhizoctonia solani*, *Fusarium sp.*, and *Phytophthora sp.*), Late Blight caused by the fungus Phytophthora infestans, Alternaria dry spot disease caused by *Alternaria solani*, bacterial wilt disease caused by *Pseudomonas bacteria*, the fungus *Peronospora parasitica*, a disease caused by a virus caused by the Tomato Yellow Net Virus (TYNV) and spot virus tomato wilt or Tomato Spotted Wilt Virus (TSWV), and a root nodule caused by root nodule nematodes (Meloidogyne spp.) [75–77].

The highest private (financial) profit of tomato farming in Pinrang Regency, South Sulawesi Province, was Rp 59,887,483 per hectare per season in the rainy season, but that in the dry season, was Rp 70,522,070 per hectare per season. It was clearly due to its higher productivity and a more competitive price in the wet season. Conversely, Kediri Regency, East Java, had the lowest private profit (financial), i.e., Rp 24,915,466 per hectare in the rainy season and Rp 27,713,132 per hectare in the dry season. It was mainly due to low productivity and a low selling price. Its relatively small farmland holding, pest and disease attacks, and low selling price as the tomatoes from this Kediri regency competing with those from the other producing areas. Tomato farming productivity is affected by inputs applied by the farmers. Limited production input availability could determine the private profit level received by the farmers in the study locations. It was in accordance with the research findings of Anang et al. (2013) [78] that tomato farming in the Wenchi region of Ghana dealt with production constraints, caused the farmers difficulty in improving productivity and earning profit, making this commodity less competitive.

Tanah Karo Regency, North Sumatera Province, had the highest economic profit (social) from tomato farming, amounting to Rp 109,565,302 per hectare per season in the rainy season, and it went up higher in the dry season, to Rp 202,874,734 per hectare per season. It was supported by its appropriate agroecology for tomato farming and good logistic access from the producing area to the Belawan Seaport. On the other hand, Kediri Regency, East Java Province, had the lowest economic profit (social profit), namely Rp 34,716,430 per hectare per season in the rainy season and Rp 43,298,910 per hectare per season in the dry season. It was mainly due to small farmland holdings, pest and disease attacks, and the relatively low selling price at the

**Table 3. Private and social profitability of tomato farming in Indonesia, 2019/2020 rainy season and 2020 dry season.**

| No | Regency | Private Profitability (Rp/Ha) | Social Profitability (Rp/Ha) |
|---|---|---|---|
| 1 | **Bandung** | | |
| | a. Rainy season | 37,833,841 | 64,143,925 |
| | b. Dry season | 58,502,797 | 67,393,233 |
| 2 | **Banjarnegara** | | |
| | a. Rainy season | 47,327,352 | 52,315,961 |
| | b. Dry season | 54,224,390 | 60,714,075 |
| 3 | **Kediri** | | |
| | a. Rainy season | 24,915,466 | 34,716,430 |
| | b. Dry season | 27,713,132 | 43,298,910 |
| 4 | **Tanah Karo** | | |
| | a. Rainy season | 56,771,963 | 109,565,302 |
| | b. Dry season | 59,806,319 | 202,874,734 |
| 5 | **Tabalong** | | |
| | a. Rainy season | 48,995,683 | 59,544,454 |
| | b. Dry season | 63,772,120 | 71,400,808 |
| 6 | **Pinrang** | | |
| | a. Rainy season | 59,887,483 | 71,923,759 |
| | b. Dry season | 70,522,070 | 83,988,576 |

farm level. Table 3 shows tomato farming profit level distribution in the research locations, i.e., private profit (financial) and economic profit (social). Attachments 3 to 14 show matrix PAM results and 13 indicators. The Java Island region consisted of (1) Bandung Regency, West Java Province; (2) Banjarnegara Regency, Central Java Province; and (3) Kediri Region, East Java Province. Off-Java Region consists of Tanah Karo Regency, North Sumatera Province, Tabalong Regency, South Kalimantan Province, and Pinrang Regency, South Sulawesi Province, and well known as the significant tomato-producing centres.

## 4.2 Competitive and comparative advantages

There are two ways of analyzing competitiveness. Specifically, financial competitiveness (private) is a competitive advantage, but economic competitiveness is a comparative advantage. The results of the analysis show that tomato farming in the study locations of the tomato-producing centres in Indonesia had a medium competitive advantage, as indicated by PCR values of 0.437 to 0.693, or less than 1, and a moderate comparative advantage, as shown by DRCR values of 0.364 to 0.616, or less than 1. PCR and DRCR coefficient values less than 1 indicated that tomato farming in both Java and off-Java regions had comparative and competitive advantages. PCR coefficient value of less than 1 shows that it needs lower domestic resource costs: those needed for producing each output value at a private price. If the DRCR coefficient is less than 1, it requires a cost that is less than the saved foreign exchange. Based on the research data, the findings showed that the comparative advantage coefficient value was higher than its competitive advantage. It was due to market and government policy distortions. Table 4 depicts consistent results in all study locations and all growing seasons. Comparatively, these research findings were not in accordance with those found by Dorogi in 2022 [52] in Spain and the Netherlands, which found that both countries had stable competitive advantages. Still, their comparative advantages were relatively weak during the study period. Table 4 reveals that PCR values were less than DRCR values in all study locations in both rainy and dry seasons. This study emphasizes the importance of strategic steps in utilizing comparative advantage

**Table 4. PCR and DRCR coefficient values of tomato farming in Indonesia, 2019/2020 rainy season and dry season, 2020.**

| No | Regency | PCR | DRCR |
|---|---|---|---|
| 1 | **Bandung** | | |
| | a. Rainy season | 0.665 | 0.526 |
| | b. Dry season | 0.554 | 0.515 |
| 2 | **Banjarnegara** | | |
| | a. Rainy season | 0.575 | 0.545 |
| | b. Dry season | 0.574 | 0.541 |
| 3 | **Kediri** | | |
| | a. Rainy season | 0.693 | 0.616 |
| | b. Dry season | 0.667 | 0.559 |
| 4 | **Tanah Karo** | | |
| | a. Rainy season | 0.525 | 0.364 |
| | b. Dry season | 0.573 | 0.435 |
| 5 | **Tabalong** | | |
| | a. Rainy season | 0.551 | 0.499 |
| | b. Dry season | 0.498 | 0.467 |
| 6 | **Pinrang** | | |
| | a. Rainy season | 0.485 | 0.436 |
| | b. Dry season | 0.437 | 0.396 |

and changing it into a competitive advantage in the global market through technological breakthroughs, efficient logistics, and effective government regulations.

Based on the coefficient values of PCR (0.667–0.693) and DRCR (0.599–0.616), Kediri Regency, East Java Province, had the lowest tomato farming competitive advantage. It was mainly due to low productivity, pest and disease attacks, and a low selling price among the competing tomato-producing centres in East Java Province. Pinrang Regency, South Sulawesi Province, had the highest competitive advantage value of PCR (0.437–0.485), but Tanah Karo Regency, North Sumatera Province, had the highest comparative advantage of DRCR (0.364–0.485). The highest competitive advantage of Pinrang Regency and the highest comparative advantage of Tanah Karo Regency were due to the relatively appropriate agroecology for tomato farming, intensive pest and disease control, high productivity, and relatively high selling price at the farm level. Based on these findings, tomato farming in the study locations was classified as having moderate competitive and comparative advantages. In general, the comparative advantages were higher than the competitive advantages, and it showed market distortions such as imperfect market mechanisms and government policy are not on the farmers' side. To improve tomato farming competitiveness, the farmers are suggested to establish a farmers' corporation such as a cooperative, to adopt smart farming, to apply marketing digitalization, to improve distribution efficiency and marketing, and to empower farmers' institutions, to connect the producing center in rural areas with the consumption center in urban areas [79, 80, 81]. Another finding reveals that tomato farming in the dry season was more profitable than in the rainy season, as there were fewer pests and disease attacks in the dry season [82]. Table 4 depicts the PCR and DRCR coefficient values for tomato farming by location and season.

## 4.3 Impact analysis of the divergence and government policy

Policy analysis synthesizes information, including research results, to produce recommendations for public policy design options [83, 84]. Public policy is a government decision or action

that affects the actions of individuals in community groups. The impact of governmental policies on the productivity of tomato cultivation in Indonesia's production hubs is being investigated.

**4.3.1 Impacts of government policy on the output sector.** The absolute measure of the impact of divergence or government policy in the PAM of tomato farming in the production centres in Indonesia consists of output transfer (OT), input transfer (IT), factor transfer (FT), and net transfer (NT). Meanwhile, the relative size is shown by several indicators, namely the nominal protection coefficient on output (NPCO), the nominal protection coefficient on input (NPCI), the effective protection coefficient (EPC), the profitability coefficient (PC), and the subsidy ratio to producer (SRP).

Table 5 reveals the OT for tomato value magnitudes and NPCO (Net Profit of Competitive Opportunity) coefficients by regency and growing season. In all study locations, OT values of tomato farming were negative except in Banjarnegara Regency, which was positive in the rainy season, i.e., Rp 1,775,000, but negative (-Rp 1,000,000) in the dry season. The biggest negative OT value was found in Tanah Karo Regency, North Sumatera, i.e., -Rp 49,486,000 in the rainy season and -Rp 16,380,000 in the dry season. In accordance with OT values, the tomato farming NPCO coefficient values in all study locations were positive (0.760–1.013), most of which were less than 1. Banjarnegara Regency, Central Java Province, had the highest NPCO of 1.013 in the rainy season and 0.994 in the dry season. The lowest NPCO value was found in Tanah Karo Regency, North Sumatera, i.e., 0.760 in the rainy season and 0.909 in the dry season. This study shows that tomato farmers in Indonesia's study locations experienced disincentive to produce this crop. Negative OT values and NPCO coefficients less than 1 indicated that the private price (financial) of tomatoes at the farm level was below its social price. The farmers coped with the oligopsonistic market structure during harvest, resulting in lower selling prices than those in the competitive market. The other disincentive policies for farmers were value-added tax (VAT) and export tax on agricultural products. It resulted in farmers' fewer efforts to enhance tomato production due to market inefficiency and unfavorable government

**Table 5. Output transfer and nominal protection coefficient on output values of tomato farming in Indonesia, 2019–2019 rainy season and 2020 dry season.**

| No | Regency | Output Transfer (OT) (Rp) | Nominal Protection Coefficient on Output (NPCO) |
|---|---|---|---|
| 1 | **Bandung** | | |
| | a. Rainy season | -21,250,000 | 0.870 |
| | b. Dry season | -7,875,000 | 0.952 |
| 2 | **Banjarnegara** | | |
| | a. Rainy season | 1,775,000 | 1.013 |
| | b. Dry season | -1,000,000 | 0.994 |
| 3 | **Kediri** | | |
| | a. Rainy season | -4,725,000 | 0.959 |
| | b. Dry season | -10,500,000 | 0.915 |
| 4 | **Tanah Karo** | | |
| | a. Rainy season | -49,486,000 | 0.760 |
| | b. Dry season | -16,380,000 | 0.909 |
| 5 | **Tabalong** | | |
| | a. Rainy season | -5,250,000 | 0.964 |
| | b. Dry season | -1,925,000 | 0.988 |
| 6 | **Pinrang** | | |
| | a. Rainy season | -4,050,000 | 0.973 |
| | b. Dry season | -6,675,000 | 0.959 |

policies on agricultural output. These study findings were in accordance with the findings of Bayzidnejad et al. (2021) and Akhtar et al. (2016) [26, 85] on OT negative values and NPCO values less than 1 on the main vegetable commodities, including tomatoes.

**4.3.2 Impacts of government policy on the input sector.** IT and NPCI demonstrate the influence of government policy on marketable inputs for tomato farming in Indonesian production centres. IT is the discrepancy between marketable production costs at private and societal pricing. IT denotes government policies affecting tradable production inputs. A positive IT value indicates government policies giving adverse incentives (taxes) on traded inputs. Government taxes cause private farmers' earnings to become lower. Meanwhile, FT values indicate the influence of disparities or government regulations on domestic issues.

In the tradable input sector and domestic factor, the government policies are trade policy (export and import tariffs), input subsidy, and value-added tax (VAT). At the same time, market distortion could raise divergence, such as through market failure. The input transfer indicator represents the gap between the tradeable input price at the private price (financial) and the production cost for tradable input at the economic price (social). NPCI measures the difference between tradable input cost and the sum of private price (financial) and tradable input cost at economic price (social). It is an important input transfer indicator. This cost discrepancy indicates that government policy in the input sector causes the private cost (financial) of tradeable input to be different from the social price of tradable input. The effectiveness parameters of tomato farming are described in detail in Table 6. Some of the IT analyses showed positive values, except those in Tanah Karo and Bandung Regencies, which were negative in the dry season.

Along with the results of IT, most of the study locations in Indonesia had NPCI values of more than 1, except those in Tanah Karo and Bandung Regencies, which were less than 1 in the dry season. These findings reveal that tomato farmers paid a higher tradable input price compared to the perfect input market mechanism. Those inputs applied by the farmers were

**Table 6. IT, NPCI, and FT values of tomato farming in Indonesia, 2019/2020 rainy season and 2020 dry season (per hectare).**

| No | Regency | Input Transfer/IT (Rp) | Nominal Protection Coefficient on Input/NPCI | Factor Transfer (Rp) |
|---|---|---|---|---|
| 1 | **Bandung** | | | |
| | a. Rainy season | 4,213,610 | 1.149 | 846,474 |
| | b. Dry season | -298,684 | 0.989 | 1,314,120 |
| 2 | **Banjarnegara** | | | |
| | a. Rainy season | 5,389,044 | 1.214 | 1,374,565 |
| | b. Dry season | 4,151,414 | 1.184 | 1,338,272 |
| 3 | **Kediri** | | | |
| | a. Rainy season | 4,655,565 | 1.190 | 420,398 |
| | b. Dry season | 3,226,673 | 1.094 | 80,667 |
| 4 | **Tanah Karo** | | | |
| | a. Rainy season | 3,226,673 | 1.094 | 80,667 |
| | b. Dry season | -15,123,746 | 0.610 | 18,861,826 |
| 5 | **Tabalong** | | | |
| | a. Rainy season | 4,574,777 | 1.174 | 723,905 |
| | b. Dry season | 4,977,742 | 1.225 | 725,946 |
| 6 | **Pinrang** | | | |
| | a. Rainy season | 7,238,636 | 1.298 | 747,641 |
| | b. Dry season | 6,978,305 | 1.297 | -186,799 |

plastic mulch, non-subsidized fertilizer, pesticides, and seed. Conversely, the positive value of domestic factor transfer indicates that the farmers paid more for the domestic factor compared to the perfect domestic factor market mechanism. These phenomena were relevant to the subsidized interest rate of the micro credit program named People Business Credit (KUR) Program. These findings of NPCI indicators are in accordance with the research findings of Bayzidnejad et al. (2021) [26] where of NPCI for some main vegetable commodities, including tomatoes in Urmia, were with a positive coefficient value of NPCI less than 1, with different magnitudes as some of them were positive with more than 1.

The results of tomato farming IT indicators in most study locations were positive in both rainy and dry seasons. In line with the IT results, the NPCI coefficient values were also positive, more than 1, indicating that the tomato farmers dealt with disincentives in the tradable input because they had to pay a higher tradable input price as the market mechanism was imperceptive. Tomato farming in Pinrang Regency, South Sulawesi Province, had the highest NPCI values of 1.298 and 1.297 in the rainy and dry seasons, respectively. However, tomato farming in Tanah Karo Regency, North Sumatera Province, had the lowest NPCI coefficient of 0.610 in the dry season and 1.094 in the rainy season. Due to the imperfect market mechanism, tomato farmers in Indonesia had to pay a higher input price. It could happen because of government policies on value-added tax (VAT) and tradable input import tariffs such as seed, fertilizer, pesticides, and mulch. On the other hand, most of the input subsidies have already been removed. The market price of non-subsidized chemical fertilizer was much higher than that of subsidized fertilizer. The COVID-19 pandemic also resulted in increased production costs for tomato farming when this study was carried out.

In all research locations, the factor transfer values were positive except in Pinrang Regency, South Sulawesi Province, which was negative. Factor transfer value in Tanah Karo Regency, North Sumatera Province, was Rp 18,861,826 per hectare in the dry season, and the lowest was -Rp 186,799 in Pinrang Regency, South Sulawesi Province. Most of the transfer factor values were positive, indicating a less competitive market structure for domestic factors and market failures such as land, labour, and credit markets in rural areas. Farmland supply was relatively fixed, and at the same time, demand for farmland kept increasing. It resulted in quite expensive farmland rent. The labour market varied among study locations. Labour supply in Java was higher than that in outside Java, and, thus, the labor wage in Java was lower. The credit market is not optimal, as only few farmers could access KUR, and most of them had to borrow commercial credit with a much higher interest rate.

**4.3.3 Impact of government policies on the input-output sector.** The divergence and government policy impacts on input and output sectors in research locations in Indonesia were indicated by Net Transfer (NT), Effective Protection Coefficient (EPC), Profitability Coefficient (PC), and Subsidy Ratio to Producer (SRP) values. EPC analysis did not take into account policy impacts on non-tradable input prices. On the contrary, NT, PC, and SRP took into account the policy impacts on tradeable input and domestic factors. Effective Protection Coefficient (EPC) is a joint analysis of the Nominal Protection Coefficient of Output (NPCO) and the Nominal Protection Coefficient of Input (NPCI). The EPC value describes the government policy on tradable input, whether it protected or discouraged effective tomato farming. Divergence analysis showed that all input and output columns were EPC was <1, NT was negative, PC<1, and SRP was negative. It indicates that tomato farmers received disincentives for producing this commodity. The farmers overcame those problems through the application of improved seed tomato quality, balanced inorganic fertilizer, fermented organic fertilizer, and mulch reuse. Table 7 depicts the research findings on the impacts of government policy and divergence on overall input and output by location and by growing season.

**Table 7. Tomato farming NT, PC, EPC, and SRP values in Indonesia, 2019/2020 rainy season and 2020 dry season.**

| No | Regency | Effective Protection Coefficient (EPC) | Net Transfer/ NT (Rp) | Profitability Coefficient (PC) | Ratio Subsidy to Producer (SRP) |
|---|---|---|---|---|---|
| 1 | **Bandung** | | | | |
| | a. Rainy season | 0.812 | -26,310,085 | 0.500 | -0.161 |
| | b. Dry season | 0.945 | -8,890,436 | 0.808 | -0.054 |
| 2 | **Banjarnegara** | | | | |
| | a. Rainy season | 0.969 | -4,988,609 | 0.905 | -0.036 |
| | b. Dry season | 0.961 | -6,489,685 | 0.893 | -0.047 |
| 3 | **Kediri** | | | | |
| | a. Rainy season | 0.896 | -9,800,964 | 0.718 | -0.085 |
| | b. Dry season | 0.846 | -15,585,777 | 0.640 | -0.125 |
| 4 | **Tanah Karo** | | | | |
| | a. Rainy season | 0.694 | -52,793,340 | 0.518 | -0.756 |
| | b. Dry season | 0.991 | -143,068,415 | 0.795 | -0.794 |
| 5 | **Tabalong** | | | | |
| | a. Rainy season | 0.917 | -10,548,772 | 0.823 | -0.073 |
| | b. Dry season | 0.948 | -7,628,688 | 0.893 | -0.049 |
| 6 | **Pinrang** | | | | |
| | a. Rainy season | 0.912 | -12,036,277 | 0.833 | -0.079 |
| | b. Dry season | 0,902 | -13,466,506 | 0.840 | -0.083 |

The EPC of tomato farming in all regencies was positive and less than 1. It reveals that domestic profit was lower than that without friction or government programs. The existing government program did not encourage farmers to produce this commodity. The highest EPC value was found in Tanah Karo Regency, Northern Sumatera Province, in the dry season, namely 0.884, which indicated significant economic changes. On the other hand, the real PC value in Tanah Karo Regency was 0.750 in the rainy season. However, the lowest EPC value was found in Kediri Regency, East Java Province, in the dry season, i.e., 0.651, and the lowest EPC in the rainy season was 0.674. The tomato farmers were disappointed with the government regulations or market distortion in the tomato market. The government regulations or market power in the input and output sectors of the tomato market made the farmers have to pay the input at a higher price and get a lower selling price for the fresh tomatoes they sold. These research findings were in line with those found by Bayzidnejad et al. (2021) [26] on the EPC coefficient values of some main agricultural commodities, including tomatoes in Urmia, with a positive coefficient value of EPC less than 1.

Net Transfer (NT) value measures the efficiency level of agricultural production. NT is the difference between the net private profit (financial) and the net economic private profit (social) of tomato farming. The NT value also describes the difference between output transfer and the sum of input tradable transfer and factor transfer. If the NT value is positive, it indicates the producer's surplus due to government policy and/or market distortion taking place at the same time in the output and input sectors. If the NT value is negative, the producer's surplus is negative. In rainy and dry seasons, NT values in Indonesia's tomato-producing centers were negative. The biggest negative NT value was found in Tanah Karo Regency, North Sumatera Province: Rp 52,793,340 per hectare in the rainy season and -Rp 143,068,415 per hectare in the dry season. It implies that the government policy on input and output sectors did not encourage the farmers to enhance tomato production. In other words, the government policy did not take the farmers' side.

PC describes the net private profit (financial) ratio to net economic profit (social). Suppose the PC value is positive and less than 1, in that case, it indicates that the existing government

policy or market intervention makes the farmers' profit lower compared to that in the perfect market mechanism or without government intervention or market intervention. If the PC value is greater than 1, the farmers get a feasible profit in the existing market. The PC coefficient in all research locations was positive and less than 1. It impliess that farmers got less profit than they received if the market was perfect or there was no market distortion. The highest PC values were found in Banjarnegara Regency, Central Java Province, namely 0.905 and 0.893 in the rainy and dry seasons, respectively. Conversely, tomato farming in Kediri Regency, East Java Province, had the lowest PC values of 0.718 and 0.640 in the rainy and dry seasons, respectively. It indicates that the government intervention caused the market to be distorted, and the profit earned by the farmers was lower due to the imperfect market.

The SRP is the incentive proportion, or net benefit, of computed income based on social costs. Negative SRP values in all study locations indicate that the government policy resulted in relatively costly tomato farming. All tomato farmers did not get subsidies but had to pay taxes, such as land, value-added (VAT), and export taxes. Positive SRP values revealed the reversed farmers' condition: tomato farming costs are relatively low, and the farmers get subsidies. The biggest negative SRP value was found in Tanah Karo Regency, North Sumatera Province, namely -0.756 in the rainy season but 0.794 in the dry season. SRP values in Banjarnegara Regency, Central Java Province, were both negative in the rainy and dry seasons, i.e., each of -0.036 and -0.047. It indicates that the government policy was detrimental to the tomato farmers because they received negative subsidies. It implies that the farmers had to pay taxes, such as land, value-added, and export taxes.

## 4.4. Sensitivity analysis

Sensitivity analysis is useful for evaluating a variable change's model behaviour [86]. Sensitivity is how the system responds to the impact taking place due to changes in certain variables, including both detrimental and beneficial ones [87]. Table 8 describes the competitive advantage and comparative advantage of tomato farming at research locations in Indonesia in response to a decline in tomato productivity or prices. PCR and DRCR are used as key indicators, where PCR and DRCR values equal to or greater than 1 indicate a loss of competitive advantage and comparative advantage in tomato farming in Indonesia. In initial conditions, Bandung Regency had PCR values of 0.554–0.665 and DRCR 0.541–0.545; Banjarnegara Regency has PCR values of 0.574–0.575 and DRCR 0.541–0.545; Kediri Regency has PCR values of 0.667–0.693 and DRCR 0.559–0.616; Tanah Karo Regency has a PCR value of 0.525–0.573 and DRCR 0.364–0.435; Tabalong Regency has PCR values of 0.489–0.551 and DRCR 0.467–0.499; and Pinrang Regency showed PCR values of 0.437–0.485 and DRCR 0.396–0.436. PCR values <1 and DRCR <1 indicate that all research locations have competitive advantages and cooperative advantages in producing tomato commodities.

Despite initially having competitive and comparative advantages, each location shows varying levels of sensitivity to changes in tomato productivity or prices. Because the percentage reduction in productivity and price of tomatoes varies in each location, the PCR value as a measure of competitive advantage in each location also fluctuates. Tomato farming in Bandung Regency, West Java, succeeded in maintaining its competitive position until the productivity or price of tomatoes decreased by 26.30% in the rainy season and 36.84% in the dry season. Banjarnegara Regency, Central Java, continues to maintain its competitive advantage even though the productivity or price of tomatoes decreases by 31.86% in the rainy season and 37.94% in the dry season. Kediri Regency, East Java, shows its competitive resilience and experiences a decline in tomato productivity or prices of 32.69% in the rainy season and 37.94% in the dry season. Tanah Karo Regency, North Sumatra, remained competitive until productivity

**Table 8. Sensitivity analysis of tomato productivity and price (PCR and DRCR = 1).**

| No. | Regency | Actual values | | Sensitivity analysis | |
|---|---|---|---|---|---|
| | | Rainy season | Dry season | Rainy season | Dry season |
| 1 | Bandung | | | | |
| | *Competitive advantage* | | | | |
| | (1) Productivity (kg/ha) | 42,500 | 45,000 | 31,322 | 28,421 |
| | (2) Price of tomatoes (Rp/kg) | 3,350 | 3,500 | 2,469 | 2,210 |
| | (3) PCR | 0.665 | 0.554 | 1.000 | 1.000 |
| | *Comparative advantage* | | | | |
| | (1) Productivity (kg/ha) | 42,500 | 45,295 | 27,154 | 27,945 |
| | (2) Price of tomatoes (Rp/kg) | 3,850 | 3,675 | 2,640 | 2,282 |
| | (3) DRCR | 0.545 | 0.541 | 1.000 | 1.000 |
| 2 | Banjarnegara | | | | |
| | *Competitive advantage* | | | | |
| | (1) Productivity (kg/ha) | 35,500 | 40,000 | 24,188 | 24,824 |
| | (2) Price of tomatoes (Rp/kg) | 4,000 | 3,850 | 2,725 | 2,482 |
| | (3) PCR | 0.575 | 0.574 | 1.000 | 1.000 |
| | *Comparative advantage* | | | | |
| | (1) Productivity (kg/ha) | 35,500 | 40,000 | 23,969 | 25,749 |
| | (2) Price of tomatoes (Rp/kg) | 3,950 | 3,875 | 2,667 | 2,494 |
| | (3) DRCR | 0.545 | 0.541 | 1.000 | 1.000 |
| 3 | Kediri | | | | |
| | *Competitive advantage* | | | | |
| | (1) Productivity (kg/ha) | 31.500 | 35,000 | 21,202 | 23,125 |
| | (2) Price of tomatoes (Rp/kg) | 3,500 | 3,250 | 2,356 | 2,147 |
| | (3) PCR | 0.693 | 0.667 | 1.000 | 1.000 |
| | *Comparative advantage* | | | | |
| | (1) Productivity (kg/ha) | 31,500 | 35,000 | 23,239 | 24,236 |
| | (2) Price of tomatoes (Rp/kg) | 3,650 | 3,550 | 2,709 | 2,458 |
| | (3) DRCR | 0.616 | 0.559 | 1.000 | 1.000 |
| 4 | Tanah Karo | | | | |
| | *Competitive advantage* | | | | |
| | (1) Productivity (kg/ha) | 43,600 | 46,800 | 30,192 | 30,620 |
| | (2) Price of tomatoes (Rp/kg) | 3,600 | 3,500 | 2,493 | 2,290 |
| | (3) PCR | 0.525 | 0.882 | 1.000 | 1.000 |
| | *Comparative advantage* | | | | |
| | (1) Yiled (kg/ha) | 43,600 | 46,800 | 20,642 | 22,216 |
| | (2) Price of tomatoes (Rp/kg) | 4,735 | 4,535 | 2,242 | 2,248 |
| | (3) DRCR | 0.364 | 0.435 | 1.000 | 1.000 |
| 5 | Tabalong | | | | |
| | *Competitive advantage* | | | | |
| | (1) Productivity (kg/ha) | 35,000 | 38,500 | 22,642 | 22,448 |
| | (2) Price of tomatoes (Rp/kg) | 4,000 | 4,000 | 2,558 | 2,332 |
| | (3) PCR | 0.551 | 0.489 | 1.000 | 1.000 |
| | *Comparative advantage* | | | | |
| | (1) Productivity (kg/ha) | 35,000 | 38,500 | 21,929 | 22,279 |
| | (2) Price of tomatoes (Rp/kg) | 4,150 | 4,050 | 2,600 | 2,344 |
| | (3) DRCR | 0.499 | 0.467 | 1.000 | 1.000 |
| 6 | Pinrang | | | | |

*(Continued)*

**Table 8.** (Continued)

| No. | Regency | Actual values | | Sensitivity analysis | |
|---|---|---|---|---|---|
| | | Rainy season | Dry season | Rainy season | Dry season |
| | *Competitive advantage* | | | | |
| | (1) Productivity (kg/ha) | 40,500 | 44,500 | 24,194 | 24,454 |
| | (2) Price of tomatoes (Rp/kg) | 3,650 | 3,500 | 2,180 | 1,923 |
| | (3) PCR | 0.485 | 0.437 | 1.000 | 1.000 |
| | *Comparative advantage* | | | | |
| | (1) Productivity (kg/ha) | 40,500 | 44,500 | 23,450 | 23350 |
| | (2) Price of tomatoes (Rp/kg) | 3,750 | 3,650 | 2,544 | 1,915 |
| | (3) DRCR | 0.436 | 0.396 | 1.000 | 1.000 |

or prices decreased by 30.75% in the rainy season and 34.57% in the dry season; Tabalong Regency, South Kalimantan, was able to maintain its competitive advantage until productivity or prices decreased by 35.31% in the rainy season and 41.69% in the dry season. Pinrang Regency, South Sulawesi, continues to maintain its competitive advantage until the productivity or price of tomatoes decreases by 40.26% in the rainy season and 45.05% in the dry season.

In the same way that the percentage reduction in productivity or price of tomatoes varies in each location, the value of DRCR as a measure of comparative advantage in each location also fluctuates. Tomato farming in Bandung Regency, West Java, succeeded in maintaining its comparative advantage position until the productivity or price of tomatoes decreased by 36.11% in the rainy season and 37.52% in the dry season. Banjarnegara Regency, Central Java, continues to maintain its comparative advantage even though the productivity or price of tomatoes decreases by 32.49% in the rainy season and by 35.63% in the dry season. Kediri Regency, East Java, shows its comparative resilience until the productivity or price of tomatoes decreases by 25.78% in the rainy season and 30.76% in the dry season. Tanah Karo Regency, North Sumatra, remains comparative until productivity decreases by 52.66% in the rainy season and 41.62% in the Tabalong Regency, South Kalimantan, was able to maintain its comparative advantage until the productivity or price of tomatoes decreased by 37.35% in the rainy season and 42.13% in the dry season, Pinrang Regency, South Sulawesi, continued to maintain its comparative advantage until the productivity or price of tomatoes decreased by 42.15% in the rainy season and 47.53% in the dry season.

This analysis highlights the unique dynamics of tomato cultivation in each location and the importance of considering production costs, productivity levels, and the price of tomatoes received by farmers. Stakeholders in tomato production centres must carefully monitor the impact of reduced levels of productivity achieved and reduced selling prices for tomatoes received by farmers. Stakeholders need to formulate strategic plans to maintain their competitive and comparative position in the global market by implementing efficient agricultural practices and integrated marketing management strategies. In this way, farmers and tomato commodity supply chain actors in the research location can continue to strive to maintain their competitive and comparative advantages even though they face ever-changing market dynamics.

## 5. Conclusion

The development of tomato farming in several production centre areas in Indonesia, in Java and outside Java, and in the rainy season and dry season, has a higher competitive advantage than the comparative advantage. Even though there are variations in the magnitude of the PCR and DRCR values between locations, and they have moderate to weak competitive and

comparative advantages, the PCR values are consistently greater than the DRCR values. It can be concluded that this tomato production system has a stable competitive advantage compared to its comparative advantage.

This research was conducted with cross-regional and cross-season coverage (not partial or case studies) to represent better farmers' and tomato business actors' biophysical and socio-economic diversity in production locations. The research results show that the financial profits obtained by tomato farmers are often smaller than the economic profits. This indicates that tomato farmers in Indonesia have a disincentive to produce tomatoes. Incentive policies are, therefore, needed to encourage farmers to plant tomatoes and continue supporting input and output market mechanisms that work well. Formulating the necessary incentive policies and, at the same time, optimizing its comparative advantage and competitive advantage are recommended to allow tomato farmers to access and benefit the facilitation of fertilizer subsidy policies. To some extent the government has implemented the policies on easing market access, and has reduced market distortions by developing efficient distribution and marketing systems through information services and a fast and accurate market. The impact of divergence brought by government policies and market input-output errors overall benefits consumers more than farmers as producers.

In addition to efforts to transform comparative advantage into competitive advantage, adequate agricultural infrastructure such as farm roads, irrigation system, and farm equipment and machinery must be widely accessible to farmers. Farmers are encouraged to use good-quality tomato seeds, complete and balanced inorganic fertilizer, appropriate organic fertilizer, and other organic and safe agricultural chemicals. Input and output markets must be accessible to farmers at competitive prices. Cold storage and chains must be available properly so fresh tomatoes can be distributed from farmers to consumers. Implementing the Integrated Agribusiness Partnership (IABP) strategy from upstream to downstream is crucial to developing tomato farming and its efficient and profitable marketing. These recommendations are from a policy strategy perspective to translate comparative advantage into competitive advantage through appropriate and fair government policies between farmers as producers and consumers as users.

The contribution to further research in the future is to analyze the competitiveness of agricultural commodities by carrying out a more complete sensitivity analysis. Researchers can also analyze the competitiveness of agricultural commodities from a green economic perspective, namely by considering the positive and negative external impacts of the production system of an agricultural commodity.

## 5.1 Limitations and potential for future research

The limitations of this study are that it does not cover all areas of tomato production centres in Indonesia. The object of research related to the competitiveness analysis of fresh tomatoes in Indonesia is still rare, and the diversity of respondents and the number of respondents that are still limited. However, the selection of tomato farmer respondents had considered the wide variation of land tenure in each location, and the data are believed to be valid. For further research, studying the same theme about the competitiveness analysis of fresh tomatoes in Indonesia is encouraged. For the research to be more comprehensive and holistic, several major commodities can be added as competitors' commodities [26]. In addition, research locations can be expanded to the new development of production centres.

## Supporting information

**S1 Appendix. Appendix 1.** Allocation of costs into components of tradable inputs and domestic factors of tomato farming in production centres in Indonesia, 2019–2020; Appendix 2.

Shadow Prices of Input and Output of Tomato Farming in Indonesia, Rainy Season, 2019–2020 (IDR/unit); Appendix 3. Shadow Prices of Tomato Farming Inputs and Outputs in Indonesia, Dry Season, 2020 (IDR/unit); Appendix 4. The Results of the Policy Analysis Matrix (PAM) Analysis of Tomato Farming in Bandung Regency, West Java, Rainy Season, 2019/2020; Appendix 5. Results of Policy Analysis Matrix (PAM) tomato farming in Bandung Regency, West Java, Dry Season, 2020; Appendix 6. Results of the Policy Analysis Matrix (PAM) for tomato farming in Banjarnegara Regency, Central Java, The rainy season, 2019/2020; Appendix 7. The Results of the Policy Analysis Matrix (PAM) for tomato farming in Banjarnegara Regency, Central Java, Dry Season, 2020; Appendix 8. Results of the Policy Analysis Matrix analysis of tomato farming in Kediri Regency, East Java, Rainy Season, 2019/2020; Appendix 9. Results of the Policy Analysis Matrix Analysis of Tomato Farming in Kediri Regency, East Java, Dry Season, 2019/2020; Appendix 10. Results of the Policy Analysis Matrix analysis of tomato farming in Tanah Karo Regency, North Sumatera, Rainy Season, 2019/2020; Appendix 11. Results of the Policy Analysis Matrix Analysis of Tomato Farming in Tanah Karo Regency, North Sumatera, Dry Season, 2020; Appendix 12. Results of the Policy Analysis Matrix analysis of Tomato Farming in Tabalong Regency, South Kalimantan, Rainy Season, 2019/2020; Appendix 13. Results of the Policy Analysis Matrix Analysis of Tomato Farming in Tabalong Regency, South Kalimantan, Dry Season, 2020; Appendix 14. Results of the Policy Analysis Matrix analysis of tomato farming in Pinrang Regency, South Sulawesi, Rainy Season, 2029/2020; Appendix 15. Results of the Policy Analysis Matrix analysis of tomato farming in Pinrang Regency, South Sulawesi, Dry Season, 2020.
(DOCX)

## Acknowledgments

The authors thank the farmer, the farmer group management, the trader respondents, and the Agriculture Service Office staff in Regency Levels for their helpful data and information for this research. We also give our gratitude to the Directorate General of Horticulture, Ministry of Agriculture Indonesia, for the collaboration of the evaluation activities of the Horticulture Commodity Development Program, especially the tomato crop.

## Author Contributions

**Conceptualization:** Saptana, Syahrul Ganda Sukmaya, Irwanda Wisnu Wardhana, Bambang Sayaka, Nyak Ilham, Elna Karmawati, Sri Hery Susilowati, Sumaryanto, Handewi Purwati Saliem.

**Data curation:** Saptana, Fadhila Dhia Malihah, Irwanda Wisnu Wardhana, Atika Dian Pitaloka, Shabrina Austin Ghaisani, Bambang Sayaka, Nyak Ilham, Elna Karmawati, Mewa Ariani, Sri Hery Susilowati, Sumaryanto, Handewi Purwati Saliem.

**Formal analysis:** Saptana, Syahrul Ganda Sukmaya, Atika Dyah Perwita, Fadhila Dhia Malihah, Irwanda Wisnu Wardhana, Bambang Sayaka, Nyak Ilham, Elna Karmawati, Sri Hery Susilowati, Sumaryanto.

**Investigation:** Saptana, Atika Dyah Perwita, Irwanda Wisnu Wardhana, Atika Dian Pitaloka, Shabrina Austin Ghaisani, Bambang Sayaka, Nyak Ilham, Elna Karmawati, Mewa Ariani, Sumaryanto, Handewi Purwati Saliem.

**Methodology:** Saptana, Syahrul Ganda Sukmaya, Atika Dyah Perwita, Fadhila Dhia Malihah, Irwanda Wisnu Wardhana, Bambang Sayaka, Nyak Ilham, Elna Karmawati, Mewa Ariani, Sri Hery Susilowati, Sumaryanto, Handewi Purwati Saliem.

**Validation:** Saptana, Syahrul Ganda Sukmaya, Atika Dyah Perwita, Fadhila Dhia Malihah, Irwanda Wisnu Wardhana, Atika Dian Pitaloka, Shabrina Austin Ghaisani, Bambang Sayaka, Mewa Ariani, Sri Hery Susilowati, Sumaryanto, Handewi Purwati Saliem.

**Writing – original draft:** Saptana, Syahrul Ganda Sukmaya, Atika Dyah Perwita, Fadhila Dhia Malihah, Atika Dian Pitaloka, Shabrina Austin Ghaisani, Mewa Ariani, Sri Hery Susilowati.

**Writing – review & editing:** Saptana, Syahrul Ganda Sukmaya, Atika Dyah Perwita, Fadhila Dhia Malihah, Atika Dian Pitaloka, Shabrina Austin Ghaisani, Bambang Sayaka, Nyak Ilham, Elna Karmawati, Mewa Ariani, Handewi Purwati Saliem.

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
