## [Decision Letter · Decision Letter 0]

6 Mar 2023

PONE-D-22-29206Competitiveness Analysis of Fresh Tomato in Indonesia: Turning Comparative Advantage into Competitive AdvantagePLOS ONE

Dear Dr. Syahrul Ganda Sukmaya
,

Thank you for submitting your manuscript to PLOS ONE. After careful consideration, we feel that it has merit but does not fully meet PLOS ONE’s publication criteria as it currently stands. Therefore, we invite you to submit a revised version of the manuscript that addresses the points raised during the review process.

We look forward to receiving your revised manuscript.

Kind regards,

Essossinam Ali, Ph.D

Academic Editor

PLOS ONE

Journal Requirements:

2. We note that Figure 1 in your submission contain map images which may be copyrighted. All PLOS content is published under the Creative Commons Attribution License (CC BY 4.0), which means that the manuscript, images, and Supporting Information files will be freely available online, and any third party is permitted to access, download, copy, distribute, and use these materials in any way, even commercially, with proper attribution. For these reasons, we cannot publish previously copyrighted maps or satellite images created using proprietary data, such as Google software (Google Maps, Street View, and Earth). For more information, see our copyright guidelines: http://journals.plos.org/plosone/s/licenses-and-copyright.

Additional Editor Comments:

Dear Syahrul Ganda Sukmaya,

Manuscript ID PONE-D-22-29206 "Competitiveness Analysis of Fresh Tomato in Indonesia: Turning Comparative Advantage into Competitive Advantage" which you submitted to PLOS ONE has been reviewed. The comments of the referee(s) are included at the bottom of this letter.

A revised version of your manuscript that takes into account the comments of the referee(s) will be reconsidered for publication.

Please note that submitting a revision of your manuscript does not guarantee eventual acceptance, and that your revision may be subject to re-review by the referee(s) before a decision is rendered.

Once again, thank you for submitting your manuscript to The PLOS ONE journal and I look forward to receiving your revision.

Yours Sincerely,

Essossinam Ali, PhD

Academic Editor

Reviewers' comments:

Reviewer's Responses to Questions

**Comments to the Author**

1. Is the manuscript technically sound, and do the data support the conclusions?

Reviewer #1: Yes

Reviewer #2: Partly

2. Has the statistical analysis been performed appropriately and rigorously? 

Reviewer #1: No

Reviewer #2: Yes

3. Have the authors made all data underlying the findings in their manuscript fully available?

Reviewer #1: Yes

Reviewer #2: Yes

4. Is the manuscript presented in an intelligible fashion and written in standard English?

Reviewer #1: No

Reviewer #2: Yes

5. Review Comments to the Author

Reviewer #1: I concent to the sampling of the study location and respondents. (1) What justified you chose 6 district for the study location?. please clarity the location related the representative of the tomato center and farmers. Is it represent or not?. (2). Then you used the FGD to data collection and only 10 farmers. I think its not enough and need more sample. (3) Please change your recommendation follow your objective and more implemented. (4) Please check the similarity index

Reviewer #2: The manuscript focuses on the actual topic - financial and economic profitability of tomato farming. Authors applied competitive and comparative advantages and assessed the impact of government policies. Finally, they suggests some strategic policies to create comparative advantage into its competitive advantage in the market. The manuscript is quite inconsistent and very extensive and meandering. On the other side, I miss detail discussion.

Authors should considers following recommendations:

I appreciate deep analyses of literature reviewed however, I have some small comments to improve the manuscript. Literature Review is extensive. Some parts should be consider as methodology, (e.g. Line 212-220) and some part are not directly interlinking with following research (e.g. Line 271-283). Often it discusses a topic that is not directly related to the research results. (e.g. Line 106-114 How is interlinking a characterisation of tomato nutritional composition with the topic of research?)

On the other side some part of Methodology could be considered as Literature Review – (Line 287-310).

Chapter “Results” also consists some theoretical descriptions, which should be move as methodology part. (e.g. Line 549-569)

In results describes claims that are speculative, without substantiated research. (e.g. Line 516: In general, the research findings suggest that the value of the competitive advantage coefficient is higher than the benefit of the comparative advantage….. Could you in detail explain sentences like that?)

On the other side, some parts (like Discussion) should be described in more detail. The authors could easily implement the discussion as a separate part of the manuscript, to compare their results with similar research (should be mentioned in Literature Review) and highlight scientific novelty of the study.

It is necessary to supplement your conclusions with the knowledge of other researches. Moreover, some of your conclusion and suggestion could be considered as speculation, because there are not directly interlinking with results. (e.g. Line 702-709)

Additionally, the conclusion must better highlighted novelty and the scientific significance of the results.

6. PLOS authors have the option to publish the peer review history of their article (what does this mean?). If published, this will include your full peer review and any attached files.

Reviewer #1: No

Reviewer #2: No

---

## [Author Response · Author response to Decision Letter 0]

19 Apr 2023

Response to reviewer and editor comments:

1. Research Location Figure:

Response to Editor Comment:

 We have remove our research location figure and change it in to word.

Response to Reviewer Comments:

1) Is the manuscript technically sound, and do the data support the conclusions?

Reviewer #1: Yes

Reviewer #2: Partly

Response: 

The total sample was 15 farmer households for each district in Java Island and 10 farmer households for each district in the Outer Java region. The sample selection of farmer households represents large land farmers, medium land farmers and small land farmers. Where farmers in each class use relatively homogeneous technology, data and information are obtained from agricultural extension workers and district agricultural officers.

Besides that, Focus Group Discussions (FGD) were also conducted at each research location which consisted of farmer group leaders and administrators, farmer/advanced farmer leaders, field agricultural extension officers, district agriculture service officers, village collectors, traders between regions, wholesalers. / wholesaler and exporter.

The conclusions have been corrected and supported by data analysis results.

2) Has the statistical analysis been performed appropriately and rigorously?

Reviewer #1: No

Response: 

This study uses a mathematical approach (Policy Analysis Matrix), not a statistical approach so it does not require a certain number of respondents (N), but emphasizes the strength of the respondents' representation. Respondents were selected purposively to represent the biophysical and socioeconomic variations in each study location.

Reviewer #2: Yes

3) Have the authors made all data underlying the findings in their manuscript fully available?

Reviewer #1: Yes

Reviewer #2: Yes

4) Is the manuscript presented in an intelligible fashion and written in standard English?

Reviewer #1: No

Response: 

The language structure has been improved to make it clearer and correct. Typical and minor errors have been corrected and refined.

Reviewer #2: Yes

5) Review Comments to the Author

Reviewer #1: I concent to the sampling of the study location and respondents. (1) What justified you chose 6 district for the study location?. please clarity the location related the representative of the tomato center and farmers. Is it represent or not?. (2). Then you used the FGD to data collection and only 10 farmers. I think its not enough and need more sample. (3) Please change your recommendation follow your objective and more implemented. (4) Please check the similarity index

Dear Reviewer #1. We thank you for your valuable comments. Please find our answers to your inquiries as the following:

1) here are six study locations selected in this research, consisting of 3 regencies on Java Island namely: (1) Bandung Regency representing the West Java production center area; (2) Banjarnegara Regency representing the Central Java Province production center area, (3) Kediri Regency representing the East Java production center area and 3 other regencies outside Java Island namely; (4) Tanah Karo Regency representing production centers in North Sumatra, (5) Tabalong Regency representing production centers in the developing South Kalimantan Province, and (6) Pinrang Regency representing production centers in South Sulawesi. BPS data shows that reflected tomato production in Java contributed 464,865.4 tons (41.71%), Sumatra Island contributed 392,964.2 tons (35.26%), Kalimantan Island contributed 27,789 tons (2.49%), and South Sulawesi contributed 156,608.5 tons (14.07%) of Indonesia's total production of 1,114,399.5 tonnes (100%). Furthermore, the locations were chosen based on whether they had exported tomatoes or had the potential to export tomatoes. As a result, the author believes that the research locations chosen are representative of the performance of tomato production in Indonesia.

2) This study did not only use the FGD method, but also a limited survey: (i) for farmer respondents, there were 15 respondent regencies in Java and 10 respondent regencies for areas outside Java Island; (ii) In this research conducted a limited survey was also carried out through interviews with traders at various levels, both collecting traders as many as 2, traders between regions 2 and wholesalers 2 and retailers 2 people; and (iii) in addition, farmer group leaders and farmer group administrator, field agricultural extension workers, field extension coordinators, Agriculture Service officers in each sample district, traders, packing house entrepreneurs, and exporters participated in FGDs, with approximately 10 FGD participants per sample district.

3) In addition to statistical analysis, this study uses mathematical analysis which does not require a certain number of samples as in statistical analysis. However, the selected sample has been attempted to be representative of each location. Thus the research sample totaling 10-15 farmer respondents and supplemented by FGDs attended by groups of farmers and farmers, extension workers, Agricultural Service officers, collectors, wholesalers, and exporters (10 respondents) have represented all farmers and business actors in the research area.

4) The similarity index was calculated, and the result is 10%.

Reviewer #2: The manuscript focuses on the actual topic - financial and economic profitability of tomato farming. Authors applied competitive and comparative advantages and assessed the impact of government policies. Finally, they suggests some strategic policies to create comparative advantage into its competitive advantage in the market. The manuscript is quite inconsistent and very extensive and meandering. On the other side, I miss detail discussion.

Authors should considers following recommendations:

I appreciate deep analyses of literature reviewed however, I have some small comments to improve the manuscript. Literature Review is extensive. Some parts should be consider as methodology, (e.g. Line 212-220) (have changed to line 429-436) and some part are not directly interlinking with following research (e.g. Line 271-283) (have been erased). Often it discusses a topic that is not directly related to the research results. (e.g. Line 106-114 How is interlinking a characterisation of tomato nutritional composition with the topic of research?) (have been added a linking sentences in line 112-116)

On the other side some part of Methodology could be considered as Literature Review – (Line 287-310) (Have been changed to Literatur Review Part (line 167-190).

Chapter “Results” also consists some theoretical descriptions, which should be move as methodology part. (e.g. Line 549-569) (have been changed in ke line 397-418)

In results describes claims that are speculative, without substantiated research. (e.g. Line 516: In general, the research findings suggest that the value of the competitive advantage coefficient is higher than the benefit of the comparative advantage….. Could you in detail explain sentences like that?) (has been corrected and added with the alignment of the results of previous studies in Spain and the Netherlands, data is also shown in table 4, at line 565-572)

On the other side, some parts (like Discussion) should be described in more detail. The authors could easily implement the discussion as a separate part of the manuscript, to compare their results with similar research (should be mentioned in Literature Review) and highlight scientific novelty of the study.

It is necessary to supplement your conclusions with the knowledge of other researches. Moreover, some of your conclusion and suggestion could be considered as speculation, because there are not directly interlinking with results. (e.g. Line 702-709) (have been erased in the main article)

Additionally, the conclusion must better highlighted novelty and the scientific significance of the results.

Dear Reviewer #2. We thank you for your detail examination on our paper. Please find our answers to your inquiries as the following:

Response:

1) Harmonized the whole section of the Literature Review, Methodology, and Results

Response:

We have harmonized the whole section of the Literature Review, Methodology, and Results as you suggested. The new body of text has improved the quality of this paper. By restructuring sentences, removing some irrelevant literature, and adding the most recent relevant literature, improvements have been made in a more systematic and focused manner.

Note: Besides the FGD there is a limited survey that needs to be explained in the methodology

2) Literature Review is extensive 

Response: 

The literature has been simplified by removing literature that is not directly related to the topic and adding literature that is related.

3) Some parts should be consider as methodology

Response:

The literature on theory and concepts has been moved to the methodology chapter and literature that is empirical review has been included in the Literature Review chapter.

4) Some part are not directly interlinking with following research (e.g. Line 271-283)

Response:

The authors agreed that non-related literature should be excluded and partially replaced with related literature and the latest.

5) Often it discusses a topic that is not directly related to the research results.

Response:

The authors agreed that non-related literature should be excluded.

6) How is interlinking a characterisation of tomato nutritional composition with the topic of research?

Response:

The connection between tomato composition characterization and competitiveness can be explained as follows: market demand conditions describe consumer demand for goods and services produced by producers. Current conditions These, global consumer demand necessitates more comprehensive and detailed product attributes such as safety, nutritional, value, attributes, ecolabel, product traceability dan humanistic.

Borrowing the term introduced (Porter, 1990) sophisticated and demanding buyers, namely consumers who are intelligent and critical will provide valuable information to producers regarding the quality of products and services needed by consumers.

7) On the other side some part of Methodology could be considered as Literature Review – (Line 287-310). (have been Changed to line 167-190)

Response:

Empirical research literature has been moved from the methodology section to the literature review section, and vice versa, the literature which is a theoretical conceptual review has been transferred to the methodology section.

8) Chapter “Results” also consists some theoretical descriptions, which should be move as methodology part. (e.g. Line 549-569) (have been Changed to metodologi part in line 397-418)

Response: 

The authors agreed to move the competitiveness literature review, which is a theoretical and conceptual review, to the methodology section.

9) In results describes claims that are speculative, without substantiated research. (e.g. Line 516: In general, the research findings suggest that the value of the competitive advantage coefficient is higher than the benefit of the comparative advantage….. Could you in detail explain sentences like that?)

Response:

These findings are not speculative because the findings of this study show consistent results across regions and seasons. Furthermore, to support these findings, the results of other researchers' studies have been reviewed in the literature. To show that this study is not speculative, it is necessary to show the figures in the table that the general level of PCR competitiveness in several locations is greater than the DRCR value.

10) On the other side, some parts (like Discussion) should be described in more detail. The authors could easily implement the discussion as a separate part of the manuscript, to compare their results with similar research (should be mentioned in Literature Review) and highlight scientific novelty of the study.

Response:

It has been completed with a relevant literature review, comparative analysis was carried out with the results of studies by other researchers, and has been synthesized. Article status and scientific updates have been completed.

11) It is necessary to supplement your conclusions with the knowledge of other researches. Moreover, some of your conclusion and suggestion could be considered as speculation, because there are not directly interlinking with results. (e.g. Line 702-709).

Response:

The writing team may consider suggestions to focus on answering research objectives and referencing the findings of other researchers' studies, but some sections are directly related to research methodology, such as the allocation of the distribution of cost components into tradable inputs and domestic factors, as well as the calculation of social prices, and private budged and social budget are the stages and calculations of PAM.

12) The conclusion must better highlighted novelty and the scientific significance of the results. 

Response:

The novelty in this research includes: (i) this research was conducted across regions and across seasons so that it can more comprehensively represent the biophysical and socioeconomic diversity of farmers and tomato business actors in the production site compared to other studies that are case studies or partial; (ii) provide novelty that consistently that the value of the DRCR (comparative advantage) coefficient is lower than the PCR (competitive advantage) coefficient value which means that the competitive advantage from the social (economic) perspective is higher than the competitive advantage from the private (financial) perspective; (iii) provide novelty to the tomato commodity system in Indonesia that the effects of divergence caused by government policies and overall input-output market errors are more favorable to consumers than producers; and (iv) provide a policy strategy perspective to realize comparative advantage into competitive advantage through appropriate and equitable government policies between producers and consumers.

Note: It is necessary to show how many areas were used in this study and what percentage of production compared to the total national tomato production. In addition, it is also necessary to add the number of other areas that are not used in research along with the percentage of production to the total national tomato production. It is necessary to add information about Bandung's share of West Java, the share of Karo Land against North Sumatra etc.

Scientific significance of research results: (i) Provide an understanding of the use of PAM analysis tools and the stages that must be carried out (allocation of tradable input components and domestic factors, calculation of social prices, farm budget private perspective and social perspective), and finally calculate PAM indicators; (ii) Provide understanding in narrating the results of profitability analysis privately (financially) and socially (economic), competitive advantages and comparative advantages as well as the impact of divergence caused by government policies and market failure; and (iii) Formulate appropriate and equitable government policy strategies not only for consumers but also for producers.

Recommendations The results of this study can be used as input to synergize and harmonize with existing policies to be used as an improvement in tomato commodity development policies nationally. 

Note: For example, as a recommendation to reduce PPH for horticultural agricultural products, improve oligopsonistic market structure, etc.

---

## [Decision Letter · Decision Letter 1]

4 Jun 2023

PONE-D-22-29206R1Competitiveness Analysis of Fresh Tomato in Indonesia: Turning Comparative Advantage into Competitive Advantage

PLOS ONE

Dear Authors,

Thank you for submitting your manuscript to PLOS ONE. After careful consideration, we feel that it has merit but does not fully meet PLOS ONE’s publication criteria as it currently stands. Therefore, we invite you to submit a revised version of the manuscript that addresses the points raised during the review process.

We look forward to receiving your revised manuscript.

Kind regards,

Essossinam Ali, Ph.D

Academic Editor

PLOS ONE

Journal Requirements:

Reviewers' comments:

Reviewer's Responses to Questions

**Comments to the Author**

1. If the authors have adequately addressed your comments raised in a previous round of review and you feel that this manuscript is now acceptable for publication, you may indicate that here to bypass the “Comments to the Author” section, enter your conflict of interest statement in the “Confidential to Editor” section, and submit your "Accept" recommendation.

Reviewer #3: (No Response)

2. Is the manuscript technically sound, and do the data support the conclusions?

Reviewer #3: Partly

3. Has the statistical analysis been performed appropriately and rigorously? 

Reviewer #3: Yes

4. Have the authors made all data underlying the findings in their manuscript fully available?

Reviewer #3: Yes

5. Is the manuscript presented in an intelligible fashion and written in standard English?

Reviewer #3: Yes

6. Review Comments to the Author

Reviewer #3: The authors have adequately responded to questions and comments of previous reviewers. Consequently, believe the manuscript amidst the corrections is favorable for publication.

Again, the manuscript has a strong literature and methodology which served as the basis of its foundation. Technically one can consider the manuscript as sound and the data duly corresponds to the conclusion made. However, the justification for the sample size is not clear.

Also, the authors may have to re-read the manuscript to correct a few omissions and incomplete sentences. For example, in paragraphs 304 and 305, "On the other hand, countries with relatively abundant and scarce labor will tend to export labor-intensive products and import capital intensive products" There is an omission in-between "abundant and and" which I believe it is "capital". Moving to paragraphs 486 and 487, "In general, economic benefits (social) were higher than financial benefits (private), profits during rainy season were lower those during dry season". I believe here also that, there is a letter omission "than" in-between "lower and those".

7. PLOS authors have the option to publish the peer review history of their article (what does this mean?). If published, this will include your full peer review and any attached files.

Reviewer #3: No

---

## [Author Response · Author response to Decision Letter 1]

8 Jul 2023

Dear Reviewer #3. We thank you for your detailed examination on our paper. Please find our answers to your inquiries as the following:

Response:

1) Is the manuscript technically sound, and do the data support the conclusions?

Reviewer #3: Partly

Response:

We have added improvements to the sampling method and conclusions section according to the reviewer's instructions. in the sampling method section we make improvements by providing a more detailed explanation, this can be seen in lines 342-354 and 365-376. then we have also made improvements to the conclusion section, this can be seen in lines 772-776.

2) The justification for the sample size is not clear.

Response:

We have added improvements to the sampling method and conclusions section according to the reviewer's instructions. In the sampling method section we make improvements by providing a more detailed explanation, this can be seen in lines 342-354 and 365-376. then we have also made improvements to the conclusion section, this can be seen in lines 772-776.

3) Re-read the manuscript to correct a few omissions and incomplete sentences.

Response:

We have corrected omissions and incomplete grammar according to the direction of the reviewer. Our fixes are on lines 25, 32-33, 74, 182-184, 272-274, and 524-525.

---

## [Decision Letter · Decision Letter 2]

4 Sep 2023

PONE-D-22-29206R2Competitiveness Analysis of Fresh Tomato in Indonesia: Turning Comparative Advantage into Competitive Advantage

PLOS ONE

Dear Dr. Sukmaya,

Thank you for submitting your manuscript to PLOS ONE. After careful consideration, we feel that it has merit but does not fully meet PLOS ONE’s publication criteria as it currently stands. Therefore, we invite you to submit a revised version of the manuscript that addresses the points raised during the review process.

The reviewer raised important issues and I agree with him

I have also the following comments on your revised version.

1. The reviewer made important comments and raised concerns about the quality of the language. Please, proofread the manuscript and edit it properly the English to meet the Journal's standard

2. Please, your findings need to be discussed. What are the implications of your results?  This needs to be clear out.

3. What makes your manuscript different from numerous papers that use similar methods on the topic? (You did similar work on patato, and  tropical fruit)

Please submit your revised manuscript by Oct 19 2023 11:59PM If you will need more time than this to complete your revisions, please reply to this message or contact the journal office at plosone@plos.org. Please include the following items when submitting your revised manuscript:A rebuttal letter that responds to each point raised by the academic editor and reviewer(s). You should upload this letter as a separate file labeled 'Response to Reviewers'.A marked-up copy of your manuscript that highlights changes made to the original version. You should upload this as a separate file labeled 'Revised Manuscript with Track Changes'.An unmarked version of your revised paper without tracked changes. You should upload this as a separate file labeled 'Manuscript'.If applicable, we recommend that you deposit your laboratory protocols in protocols.io to enhance the reproducibility of your results. Protocols.io assigns your protocol its own identifier (DOI) so that it can be cited independently in the future. For instructions see: https://journals.plos.org/plosone/s/submission-guidelines#loc-laboratory-protocols. Additionally, PLOS ONE offers an option for publishing peer-reviewed Lab Protocol articles, which describe protocols hosted on protocols.io. Read more information on sharing protocols at https://plos.org/protocols?utm_medium=editorial-email&utm_source=authorletters&utm_campaign=protocols.

We look forward to receiving your revised manuscript.

Kind regards,

Essossinam Ali, Ph.D

Academic Editor

PLOS ONE

Journal Requirements:

Additional Editor Comments:

Dear Authors,

Sorry for the delay in deciding on your manuscript. It is sometimes difficult to find reviewers for the evaluation of the manuscript.

I have the following comments on your revised version.

1. The reviewer made important comments and raised concerns about the quality of the language. Please, proofread the manuscript and edit it properly the English to meet the Journal's standard

2. Please, your findings need to be discussed. What are the implications of your results? This needs to be clear out.

3. What makes your manuscript different from numerous papers that use similar methods on the topic?

Reviewers' comments:

Reviewer's Responses to Questions

**Comments to the Author**

1. If the authors have adequately addressed your comments raised in a previous round of review and you feel that this manuscript is now acceptable for publication, you may indicate that here to bypass the “Comments to the Author” section, enter your conflict of interest statement in the “Confidential to Editor” section, and submit your "Accept" recommendation.

Reviewer #4: (No Response)

2. Is the manuscript technically sound, and do the data support the conclusions?

Reviewer #4: Partly

3. Has the statistical analysis been performed appropriately and rigorously? 

Reviewer #4: Yes

4. Have the authors made all data underlying the findings in their manuscript fully available?

Reviewer #4: Yes

5. Is the manuscript presented in an intelligible fashion and written in standard English?

Reviewer #4: No

6. Review Comments to the Author

Reviewer #4: 1. Please avoid unnecessary explanations such as “Tomato (Lycopersicon esculentum mill) is a widely known vegetable in Indonesia” as this is not important in the context of the manuscript.

2. The manuscript needs to be edited by a professional English editor.

3. Need more rationales in the introduction section, why it is important to assess the comparative and competitive advantages.

4. Please revise the abstract to make it more interesting align with the findings and recommendations and explain more about the samples in the abstract.

5. Explore more about the findings, not only about the number but also the critical analysis. If a tomato has comparative and competitive advantages, so what? What is the rationale behind it, and what does it imply? what is the next?

6. The recommendations are not sound, please make them clear and more applicable.

7. The conclusion is different from a summary, so please refine it.

8. We found that the authors previously wrote similar ideas and topics, but in different commodities, so please explain the differences and the novelty, not only in different commodity.

7. PLOS authors have the option to publish the peer review history of their article (what does this mean?). If published, this will include your full peer review and any attached files.

Reviewer #4: No

---

## [Author Response · Author response to Decision Letter 2]

18 Oct 2023

Dear reviewer,

We have revised our article as per your suggestions. We hope our revisions meet your requirements and our article can be published. This is our responds:

Responses to 3rd Review results: 

1. Please avoid unnecessary explanations such as “Tomato (Lycopersicon esculentum mill) is a widely known vegetable in Indonesia” as this is not important in the context of the manuscript.

Response: 

It is already edited (page #1, row #25).

2. The manuscript needs to be edited by a professional English editor.

Response: 

The manuscript has been edited by the Professional English Editor at ILP (International English Program), the certificate is attached.

3. Need more rationales in the introduction section, why it is important to assess the comparative and competitive advantages. 

Response: 

Tomato is an important horticultural commodity. It is an exported commodity and Indonesia is net tomato exporter. Analyzing its competitiveness based on comparative and competitive advantages is urgent for the purpose of utilizing domestic resources efficiently, especially for promoting its export market 

The novelty of this study consists of (i) competitiveness of the tomato production system analyzed through its comparative and competitive advantages in the production centers by location, i.e., in Java and outside Java, and by growing season, i.e., rainy and dry seasons; (ii) this study analyzed the competitiveness sensitivity based on productivity variables and tomato price; (iii) the policy strategy to comprehend its comparative advantage into a competitive advantage through more efficient use of domestic resources. 

4. Please revise the abstract to make it more interesting align with the findings and recommendations and explain more about the samples in the abstract.

Response: 

Abstract has been revised in terms of its systematic writing and grammar such that it will be more interesting to the readers. The sample respondents have been added into the abstract (page #1: rows #25-40)

5. Explore more about the findings, not only about the number but also the critical analysis. If a tomato has comparative and competitive advantages, so what? What is the rationale behind it, and what does it imply? what is the next?

Response: 

The findings of the parameters’ coefficients have been explained in more detail. The followings are the detailed descriptions.: 

4.1. Private and Social Profitability (pages #13-14, paragraph #2, rows #580-606). 

4.2. Competitive and Comparative Advantages (pages #15 in paragraph #1, rows #: 612-630, and rows #631-649). 

4.3.1 Impacts of Government Policy on the Output Sector (page #16, paragraph #1, rows #658-682. 

4.3.2 Impacts of Government Policy on the Input Sector: pages #17-19, page #17, paragraph #2 rows #691-701; page #18, paragraphs #1-2, rows: 717-734; page #19, rows: 735-739

4.3.3 Impact of Government Policies on the Input-Output Sector, pages #19-20, page #19 paragraph #1 rows #: 741-754, page #20 paragraph #1 rows #760-772, page #20 paragraph #2 rows #773-783), paragraph 2 rows #784-795.

4.4 Sensitivity analysis, page #21 paragraph #1 rows #808-821), and paragraph #2 rows #822-837, Table 8, page #21 row #838.

6. The recommendations are not sound, please make them clear and more applicable.

Response: 

An applicable recommendation has been added clearly to the Conclusion. In addition, it includes the contribution for future research.

7. The conclusion is different from a summary, so please refine it.

Response: 

The conclusion has been improved by those depicted in the abstract (pages #23; rows #866-900).

8. We found that the authors previously wrote similar ideas and topics, but in different commodities, so please explain the differences and the novelty, not only in different commodities.

Response: 

Competitive advantage (PCR) and comparative advantage (DRCR) analyses were employed as tools useful for the assessment of various commodities, locations, and growing seasons. Indonesia, as the tomato net exporter, could utilize this study which is valuable to assess this commodity competitiveness specifically for policy formulation on tomato development in the study locations. To differentiate from the previous papers, the research team has added a subchapter 4.4, i.e., sensitivity analysis of productivity and price variables on competitive advantage (PCR) and comparative advantage (DRCR) (pages #21-22 rows #808-863). 

Kind regards, 

Syahrul Ganda Sukmaya

---

## [Decision Letter · Decision Letter 3]

14 Nov 2023

Competitiveness Analysis of Fresh Tomatoes in Indonesia: Turning Comparative Advantage into Competitive Advantage

PONE-D-22-29206R3

Dear Mr. Syahrul Ganda Sukmaya,

We’re pleased to inform you that your manuscript has been judged scientifically suitable for publication and will be formally accepted for publication once it meets all outstanding technical requirements.

Kind regards,

Essossinam Ali, Ph.D

Academic Editor

PLOS ONE

Additional Editor Comments (optional):

Reviewers' comments:

Reviewer's Responses to Questions

**Comments to the Author**

1. If the authors have adequately addressed your comments raised in a previous round of review and you feel that this manuscript is now acceptable for publication, you may indicate that here to bypass the “Comments to the Author” section, enter your conflict of interest statement in the “Confidential to Editor” section, and submit your "Accept" recommendation.

Reviewer #4: All comments have been addressed

2. Is the manuscript technically sound, and do the data support the conclusions?

Reviewer #4: Yes

3. Has the statistical analysis been performed appropriately and rigorously? 

Reviewer #4: Yes

4. Have the authors made all data underlying the findings in their manuscript fully available?

Reviewer #4: Yes

5. Is the manuscript presented in an intelligible fashion and written in standard English?

Reviewer #4: Yes

6. Review Comments to the Author

Reviewer #4: all of reviewer comments have been addressed and technically sounds. I recommended the manuscript to be accepted

7. PLOS authors have the option to publish the peer review history of their article (what does this mean?). If published, this will include your full peer review and any attached files.

Reviewer #4: No

---

## [Editor Report · Acceptance letter]

23 Nov 2023

PONE-D-22-29206R3 

Competitiveness Analysis of Fresh Tomatoes in Indonesia: Turning Comparative Advantage into Competitive Advantage 

Dear Dr. Sukmaya:

I'm pleased to inform you that your manuscript has been deemed suitable for publication in PLOS ONE. Congratulations! Your manuscript is now with our production department. 

Kind regards, 

on behalf of

Dr. Essossinam Ali 

Academic Editor

PLOS ONE